# Mesenchymal stromal cell-derived septoclasts resorb cartilage during developmental ossification and fracture healing

Kishor K. Sivaraj [1], Paul-Georg Majev [1], Hyun-Woo Jeong [1], Backialakshmi Dharmalingam [1], Dagmar Zeuschner [2], Silke Schröder[1], M. Gabriele Bixel[1], Melanie Timmen[3], Richard Stange [3] & Ralf H. Adams [1]✉

Developmental osteogenesis, physiological bone remodelling and fracture healing require removal of matrix and cellular debris. Osteoclasts generated by the fusion of circulating monocytes degrade bone, whereas the identity of the cells responsible for cartilage resorption is a long-standing and controversial question. Here we show that matrix degradation and chondrocyte phagocytosis are mediated by fatty acid binding protein 5-expressing cells representing septoclasts, which have a mesenchymal origin and are not derived from haematopoietic cells. The Notch ligand Delta-like 4, provided by endothelial cells, is necessary for septoclast specification and developmental bone growth. Consistent with the termination of growth, septoclasts disappear in adult and ageing bone, but re-emerge in association with growing vessels during fracture healing. We propose that cartilage degradation is mediated by rare, specialized cells distinct from osteoclasts. Our findings have implications for fracture healing, which is frequently impaired in aging humans.

---

[1] Max Planck Institute for Molecular Biomedicine, Department of Tissue Morphogenesis, and University of Münster, Faculty of Medicine, D-48149 Münster, Germany. [2] Electron Microscopy Unit, Max-Planck-Institute for Molecular Biomedicine, D-48149 Münster, Germany. [3] Department of Regenerative Musculoskeletal Medicine, Institute of Musculoskeletal Medicine (IMM), University Hospital Münster, 48149 Münster, Germany. ✉email: ralf.adams@mpi-muenster.mpg.de

In endochondral ossification during development and regeneration, avascular cartilage serves as a template that is gradually replaced by newly formed bone[1,2]. The resorption of cartilaginous matrix and hypertrophic chondrocytes is essential for the invasion of growing blood vessels and initiation of ossification. In development, failure of these processes results in the pathological disorganization of growth plate chondrocytes and shortening of long bones, which are hallmarks of chondrodysplasias, a very large and heterogeneous group of human developmental disorders affecting bone growth[3,4]. The importance of chondrocyte resorption extends into adult life, where the process is indispensable for bone fracture healing and regenerative ossification[1,2,5,6]. Cartilage resorption has been attributed to multiple different cell types including osteoclasts, which are large and multinucleated cells responsible for the degradation of mineralized bone[4,7]. It also has been suggested that osteoclasts are closely related to another population of polyploid cells found near chondrocytes, namely chondroclasts[8,9]. Recently, the production of matrix metalloproteinases (MMPs) mediating cartilage resorption was attributed to angiogenic endothelial cells (ECs) in proximity of the growth plate[10]. A few studies have credited cartilage resorption to an elusive cell population termed septoclasts (SCs), which are mononucleated, located at the chondro-osseous border in developing bone, and rich in the cysteine proteinase cathepsin B and fatty acid-binding protein 5 (FABP5)[5,11–14].

Here, we have used high-resolution imaging in combination with mouse genetics and single cell RNA sequencing (scRNA-seq) to systematically investigate the processes during developmental and regenerative osteogenesis at cellular resolution. We provide insight into the behaviour of stromal cells and show that cartilage resorption is mediated by a small population of mesenchymal cell-derived FABP5+ cells representing septoclasts.

## Results

### Morphological characterization of resorptive cells at the chondro-osseous border

To gain insight into the resorptive processes at the growth plate, we performed immunostaining of sections from 3-week-old murine femur with a range of markers and found that FABP5+ cells are mainly located at the chondro-osseous border zone (Fig. 1a, b and Supplementary Fig. 1a–c). FABP5+ cells are associated with the ECs of distal vessel buds, the leading edge of growing vasculature[15,16], and might therefore represent SCs. Notably, these FABP5+ cells express well-established markers of perivascular mesenchymal cells such as platelet-derived growth factor receptor β (PDGFRβ) and PDGFRα (Fig. 1c and Supplementary Fig. 1d). Bone-resorbing, multinucleated osteoclasts, which are marked by co-expression of CD68 and V-type proton ATPase subunit B (vATPase), are widely distributed throughout the metaphysis and can be seen at the chondro-osseous border zone but lack FABP5 expression (Fig. 1d and Supplementary Fig. 1e–g). Furthermore, FABP5+ cells are significantly more abundant than osteoclasts at the chondro-osseous interface (Fig. 1d). Electron microscopy (EM) of the femoral metaphysis shows close association of an unusual cell population, presumably representing septoclasts, with small vessels. Morphologically, these cells are spindle-shaped, elongated and mononuclear, exhibit strong actin filaments, numerous vesicular structures and an enlarged rough endoplasmic reticulum, a feature that is also seen in plasma cells where it is associated with high levels of protein production[17] (Fig. 1e and Supplementary Fig. 2a–c). The cells make physical contact with hypertrophic chondrocytes and bone cells, osteoclasts and ECs (Fig. 1e), but, unlike pericytes, they are not embedded in the subendothelial basement membrane. Immunostaining shows that PDGFRα+ and PDGFRβ+ presumptive septoclasts express the proteogylocan NG2, a marker of pericytes and other mesenchymal cell populations, but lack expression of CD146/MCAM, which is another marker of pericytes and vascular smooth muscle cells (Supplementary Fig. 2d, e). Taken together, the ultrastructural features presented above, FABP5 expression and their localization at the chondro-osseous border are consistent with septoclasts[5,11–13], whereas the expression of PDGF receptors and NG2, markers of many mesenchymal cell populations, indicates the need for further characterization.

EdU labelling indicates that FABP5+ cells are highly proliferative during early postnatal long bone formation (Supplementary Fig. 2f). During fetal bone development at embryonic day (E) 16.5 to 17.5, FABP5+ cells are mainly located at the vascular front in the primary ossification centre (POC), whereas CD68+ elongated OCs are found throughout the POC without accumulation at the edge of the growth plate (Fig. 1f and Supplementary Fig. 2g–i). FABP5+ cells are predominantly detected at the vascular front at the distal and proximal end of the metaphysis in close proximity of growth plate chondrocytes and progressively increase in number up to day 21 of postnatal long bone development (Fig. 1g and Supplementary Fig. 3a–c). The calvaria is a flat bone formed by intramembranous ossification in the absence of chondrocytes. Notably, FABP5+ cells are rare in 3-week-old calvaria and are associated with vessels penetrating the cranial bone (Supplementary Fig. 3d, e). These data indicate that FABP5+ cells in bone are predominantly associated with endochondral osteogenesis and represent a dynamic cell population at the chondro-osseous interface with characteristic properties distinguishing them from osteoclasts (Fig. 1h).

### Molecular properties and differentiation of FABP5+ cells

The origin and molecular properties of FABP5+ cells, presumably septoclasts, which represent a very rare cell population in postnatal long bone, remains to be elucidated. We, therefore, performed single cell RNA-sequencing (scRNA-seq) using the 10× Genomics Chromium platform. Making use of PDGFRα expression by the FABP5+ population, we isolated cells expressing enhanced green fluorescent protein (GFP) by fluorescence-activated cell sorting (FACS) from the metaphysis-epiphysis region of 3-week-old Pdgfra-GFP reporter mice[18] (Fig. 2a and Supplementary Fig. 4a). Droplet based scRNA-seq (see 'Methods') yielded 11,242 GFP+ cells belonging to several distinct populations (Fig. 2b and Supplementary Fig. 4b). GFP+ cells include bone mesenchymal stromal cells (BMSCs) and osteoblastic cells (OBs), which can be distinguished by the expression of Pdgfrb and Col22a1, but also Acan+ chondrocytes and S100a4+ fibroblast (Fig. 2b, c and Supplementary Fig. 4c). Reclustering of the BMSC and OB populations (7857 cells) identifies 6 sub-clusters, namely diaphyseal mesenchymal stromal cells 1 (dpMSCs1; #2382 cells) and dpMSCs2 (#1659 cells) expressing Pdgfrb and Esm1, metaphyseal mesenchymal stromal cells (mpMSCs; #1642 cells) expressing Pdgfrb and Postn, Col22a1+ OBs (#1629 cells), Top2a+ and Mki67+ proliferating BMSCs (p-BMSCs; #302 cells), and a small population of cells (#243 cells) expressing Fabp5+ and the matrix metalloproteinase Mmp9 (Fig. 2b, c and Supplementary Fig. 4d). Trajectory analysis of BMSC and OB populations indicates three separate directional branches containing dpMSCs1 and dpMSCs2, OBs, and p-BMSCs cells, respectively, whereas mpMSCs form the focal centre (Fig. 2d, e). Strikingly, Fabp5+ putative SCs remain close to mpMSCs and, as pseudotime gene expression analysis suggests, might actually arise from this population (Fig. 2d–f and Supplementary Fig. 4e). To investigate this question, we used a

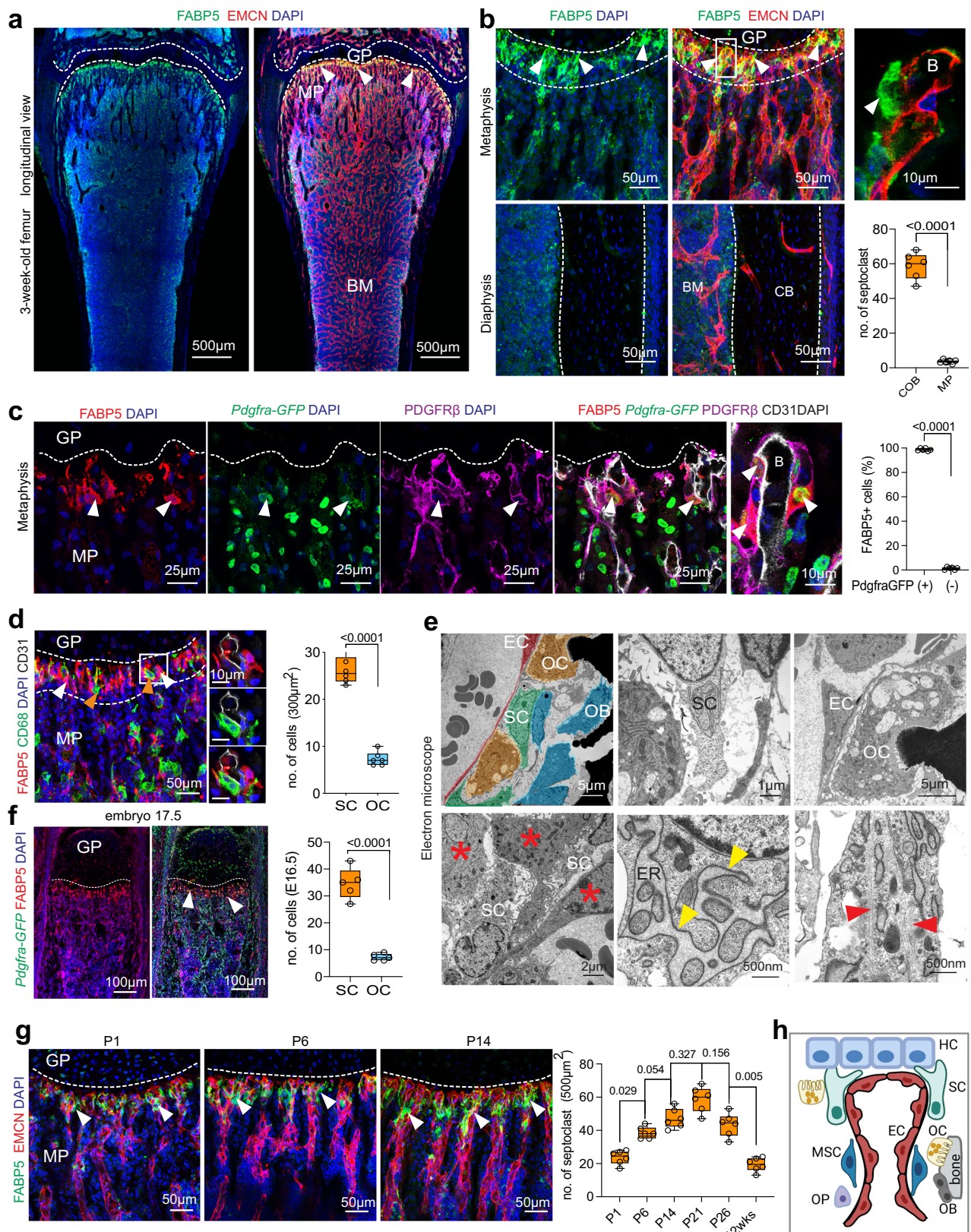

combination of immunostaining and genetic fate tracking with tamoxifen-inducible *Pdgfrb-CreERT2* line, which, in combination with the *Rosa26-mT/mG* Cre reporter[19], irreversibly labels PDGFRβ+ BMSCs by expression of GFP. Analysis of the targeted GFP+ cells at P21 shows labelling of FABP5+ cells (Fig. 2g and Supplementary Fig. 5a, b). In contrast, FABP5+ putative SCs are not labelled in *Vav1-Cre R26-mTmG* mice, which show widespread GFP expression in haematopoietic cells and therefore also in CD68+ osteoclasts (Fig. 2h and Supplementary Fig. 5c–f). These data indicate that FABP5+ cells at the chondro-osseous border, unlike osteoclasts, are not derived from the haematopoietic lineage and, instead, emerge from PDGFRβ+ metaphyseal

**Fig. 1 Characterization of SCs. a, b** Tile scan confocal longitudinal view of 3-week-old wild-type femur with FABP5+ (green) SCs around distal vasculature (EMCN, red) near the growth plate (GP, dashed lines) (**a**). SCs associated with distal EC buds (B) (arrowheads) at chondro-osseous border (COB) (dashed lines). Image on the right show higher magnifications of boxed area (**b**). Enrichment of FABP5+ cells is not seen near or inside cortical bone (CB) in the diaphysis (bottom). BM, bone marrow. Quantification shows number of SCs in COB compared to metaphysis (MP). (n = 6). **c** FABP5+ SCs (red; arrowheads) associated with CD31+ vessels (grey) at the COB show expression of *Pdgfra-GFP* reporter (green) and PDGFRβ+ (magenta). The image at higher magnification on the right depicts FABP5+ cells with nuclear GFP expression (arrowheads) in association with distal EC bud (B). Graph shows percentage of *Pdgfra-GFP*+ cells among SCs (n = 6). **d** FABP5+ SCs (white arrowheads, red) are concentrated near the growth plate (GP, dashed line), whereas CD68+ (orange arrowheads, green) osteoclasts (OCs) are widely distributed throughout 3-week-old femoral metaphysis. Quantification of SCs relative to OCs around vascular buds (n = 6). **e** Electron micrographs of metaphysis showing SCs, ECs, osteoblastic cells (OB), and OCs, as indicated. Cells in top left image are false coloured. The endoplasmic reticulum (ER) of SCs is enlarged (yellow arrowheads) and covered by ribosomes, whereas other cells (asterisks) have narrow ER cisternae. SCs are rich in actin filaments (red arrowhead, bottom right panel). (n = 3). **f** Confocal image of E17.5 femur showing *Pdgfra-GFP* (green) reporter and FABP5+ SCs (arrowheads, red) near growth plate (gp, dashed line). Quantification of E16.5 in graph on the right (n = 5). **g** Increase of FABP5+ SCs (green, arrowheads) near growth plate (GP) at P1, P6, and P14. Quantification in graph on the right shows changes in septoclasts during postnatal development (n = 5; Statistical analysis performed using Tukey multiple comparison test (one-way Anova). ECs, EMCN (red); nuclei, DAPI (blue). **h** Schematic representation of SCs, ECs in vessel buds, OCs, osteoblastic cells (OBs), mesenchymal stromal cells (MSC), and hypertrophic chondrocytes (HC) at the chondro-osseous border. n = biological independent samples and data are presented as mean values ± SEM. **b–d**, **f** Statistical analysis performed using Mann–Whitney test (two-tailed). Source data are provided in a Source data file.

mesenchymal stromal cells. Given that the FABP5+ population exhibits major hallmarks of septoclasts, our results indicate that the latter are derived from BMSCs and are thereby distinct from osteoclasts, which emerge from haematopoietic cells.

**Function of septoclasts in long bone formation**. The scRNA-seq data provides insight into the biological function and molecular properties of FABP5+ putative SCs (Fig. 3a). Gene ontology (GO) analysis shows extracellular matrix organization, extracellular matrix disassembly, and proteolysis as top biological processes for these cells (Supplementary Fig. 6a). Cellular pathway analysis reveals that genes relating to MMPs, endochondral ossification, and focal adhesion signalling pathway are also enriched (Fig. 3b). Interestingly, enriched MGI mammalian phenotypes include terms such as increased width of hypertrophic chondrocyte zone, decreased angiogenesis, and abnormal vascular and bone morphology (Supplementary Fig. 6b). The analysis of scRNA-seq data also shows that the protease genes *Mmp9*, *Mmp13*, *Mmp14*, *Mmp11*, *Ece1*, and *Adam19*, and *Ctsb* (encoding cathepsin B) are highly expressed in the *Fabp5*+ population, arguing that it indeed represents resorptive cells and therefore septoclasts (Fig. 3c and Supplementary Fig. 6c). Other highly expressed genes are related to Rho family GTPases (*Rhod*, *Rhoj*), the cytoskeleton (*Tubb2a*, *Myh9*, *Vasp*, *Actn1*, *Tln1*, *Flna*), and cell-matrix interactions (*Fn1*, *Lama4*, *Itgb5*, *Itgav*) (Supplementary Fig. 6c). Consistent with the findings above, immunosignal for MMP9, an important regulator of angiogenesis and osteogenesis[20,21], decorates FABP5+ cells in the chondro-osseous border zone as well as OCs in the same region (Fig. 3d, e and Supplementary Fig. 6d, e). Similarly, MMP13 and MMP14 are also concentrated in the chondro-osseous border zone containing FABP5+ SCs. Low anti-MMP13 signals mark also other cells in the postnatal metaphysis (Fig. 3e and Supplementary Fig. 6e–g). Previous studies have shown that the inactivation of genes encoding matrix metalloproteinases, namely *Mmp9* (ref. [20]), *Mmp13* (ref. [22]), *Mmp14* (ref. [23]), and the metallopeptidase *Adam19* (ref. [24]) result in the expansion of hypertrophic chondrocytes and impaired endochondral ossification[25]. Consistent with the known role of MMP9 and MMP13 in the degradation of cartilage collagen and aggrecan[22], MMP9 immunosignals decorate the surface of hypertrophic chondrocytes (Fig. 3f), which will undergo apoptosis to enable bone growth. Analysis of *Acan-Cre*-labelled, GFP-expressing chondrocytes at the chondro-osseous interface shows GFP+ debris inside FABP5+ and MMP9+ SCs but not inside CD68+ osteoclasts in the same area (Fig. 3g and Supplementary Fig. 6h). Furthermore, FABP5+ SCs

show strong immunostaining for lysosomal-associated membrane protein 1 (LAMP1), a major component of lysosomes mediating the breakdown of biomolecules. In contrast, osteoclasts at the chondro-osseous border show comparably little LAMP1+ signal (Fig. 3h). This is supported by electron microscopy in combination with immunogold single and double labelling of LAMP1 and vATPase (see 'Methods'). SCs stain positive for LAMP1 but are negative for vATPase, whereas the opposite applies to osteoclasts (Fig. 3i and Supplementary Fig. 6i). In addition, when septoclasts and chondrocytes (CHO) are cultured in direct proximity in vitro, SCs show strong MMP9 expression at the SC-CHO border zone (Supplementary Fig. 7a, b). Analysis of sorted cells from these co-cultures show that *Fabp5*, *Mmp9* and *Mmp14* transcripts are primarily found in SCs, whereas Acan+ shows the expected CHO enrichment and *Mmp13* expression is not confined to one cell type (Supplementary Fig. 7c, d). SCs display strong cell shape changes, extend numerous protrusions, show stronger staining of the actin cytoskeleton, and are able, to some extent, to remove calcium phosphate from pre-coated tissue culture surfaces (Supplementary Fig. 7e, f).

Further arguing that SCs secrete MMPs and thereby degrade cartilage matrix and phagocytose hypertrophic chondrocytes, the high abundance of FABP5+ and MMP9+ SCs in development mirrors active growth plate remodelling when hypertrophic chondrocytes are degraded during endochondral ossification (Supplementary Fig. 7g, h). In contrast, SCs and MMP9 expression are reduced in adult long bone and immunosignals are absent in aging animals where the growth plate has been converted into an epiphyseal line (Fig. 3j and Supplementary Fig. 7i, j). Together the data above demonstrate that FABP5+ SCs are distinct from classical osteoclasts and participate in cartilage degradation by providing secreted proteases and by phagocytosis of chondrocyte fragments (Fig. 3k).

**Endothelial–septoclast interactions**. To gain insight into the interactions between SCs and surrounding cells, we spatially separated the metaphysis region from 3-week-old wild-type long bone and performed scRNA-seq after depletion of haematopoietic lineage and progenitor cells (see 'Methods') (Fig. 4a). We sequenced 4386 metaphyseal cells, which were clustered and identified as ECs, BMSCs and OBs according to gene expression (Fig. 4b–d). Next, we merged this dataset and the *Pdgfra-GFP* scRNA-seq results (see 'Methods') before SC, mpMSC, OB and EC populations were extracted from the merged dataset for further analysis (Fig. 4e and Supplementary Fig. 8a–c). Cluster-specific expression of marker genes identified 3 distinct EC sub-

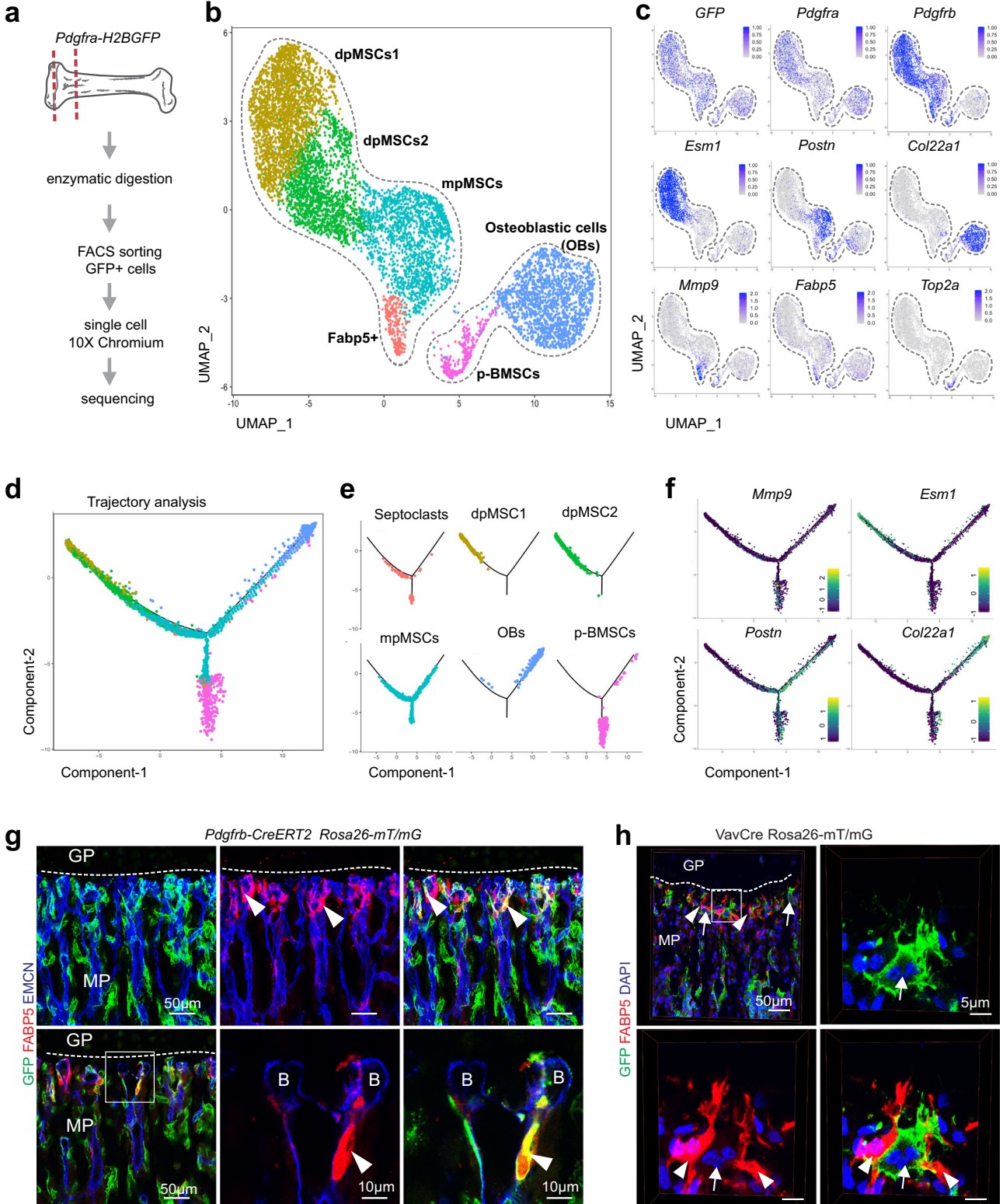

**Fig. 2 Molecular signature and differentiation of SCs. a** Preparation of 3-week-old *Pdgfra-GFP* metaphysis-epiphysis region from femur and tibia for scRNA-seq analysis. **b, c** scRNA-seq of 3-week-old *Pdgfra-H2BGFP* reporter bone. UMAP plot of colour-coded cell clusters within *Pdgfra-GFP*+ positive cells (**b**) and visualization of cell type-specific marker genes (**c**). **d–f** Monocle trajectory analysis of BMSC lineage differentiation with coloured cell clusters (**d**) and representation of individual cluster in trajectory (**e**). Cell type-specific relative gene expression shown in pseudo-time (**f**). **g, h** Representative confocal images of 3-week-old *Pdgfrb-CreERT2 R26-mT/mG* reporter femur with GFP+ (green) and FABP5+ (red) SCs (arrowheads) near vessel buds (B). Bottom panels show a single optical plane (left) and higher magnifications of the boxed area (centre and right), respectively (**g**). *Vav1-Cre*-controlled *R26-mT/mG* reporter activation (GFP expression) labelling hematopoietic cells and osteoclasts (arrow) but not FABP5+ (red) SCs (arrowheads) (**h**). Independent animals, **g, h** (*n* = 4).

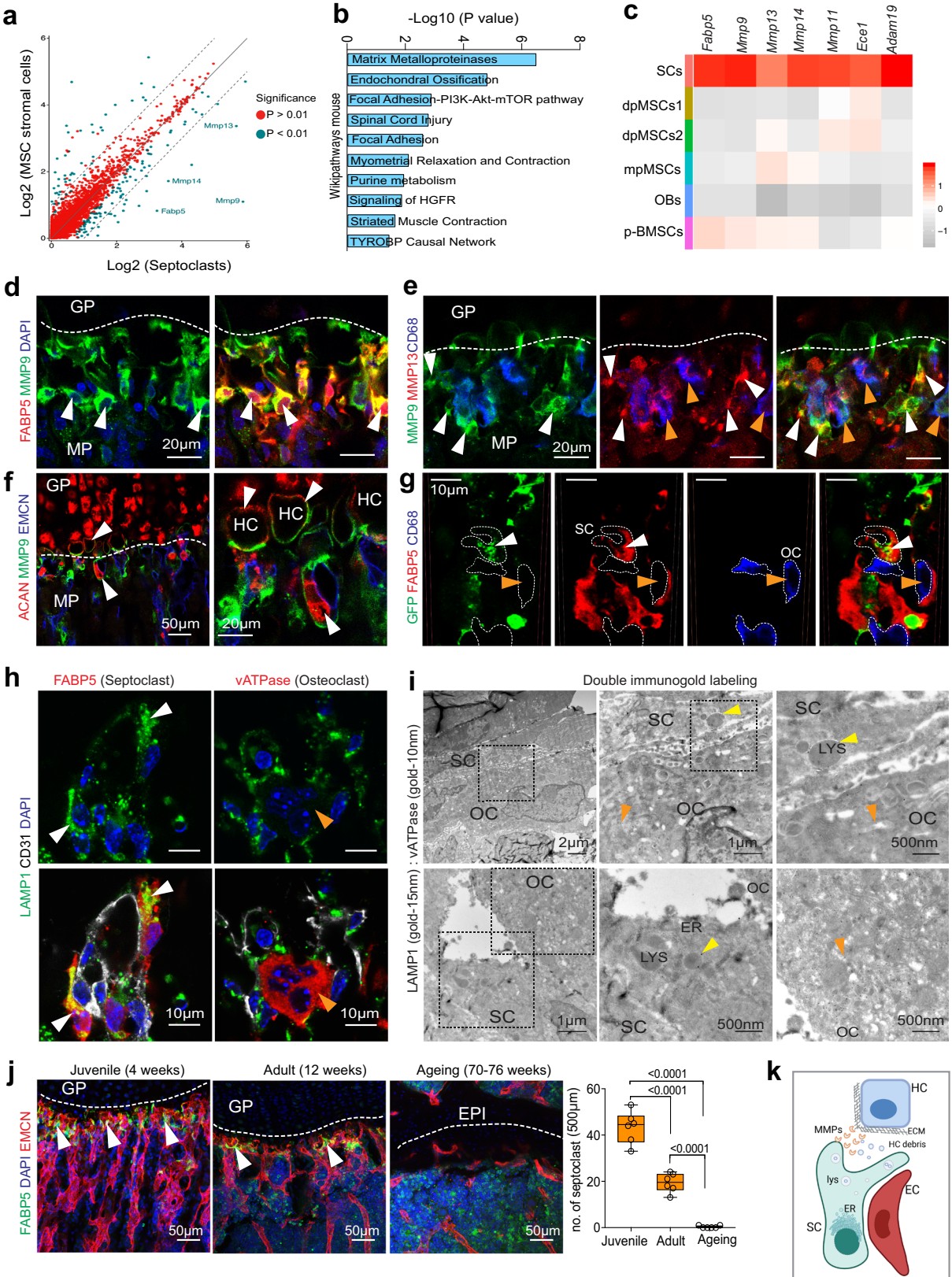

clusters, namely arterial ECs (aECs), metaphyseal ECs (mpECs) resembling the previously published CD31[high] EMCN[high] (type H) endothelial subpopulation[16], and bone marrow ECs (bmECs) of the sinusoidal vasculature in the BM cavity in addition to *Fabp5*+ SCs, mpMSCs and OBs (Supplementary Fig. 8d, e). Confirming our previous results, the matrix metalloproteinases

*Mmp9, Mmp13, Mmp14,* and *Mmp11* are highly expressed by *Fabp5*+ SCs relative to other populations (Fig. 4f). We also observed that the Notch pathway genes Hairy/enhancer-of-split related with YRPW motif protein 1 (*Hey1*) and *Hey-like* (*Heyl*) are highly expressed by SCs (Fig. 4g). Transcripts for the Notch ligands *Dll1, Dll4, Jag1, Jag2* are predominantly expressed in ECs,

**Fig. 3 Septoclast function in bone. a** Scatter plot showing differential gene expression (DGE) in SCs relative to other bone mesenchymal stromal cells. DGE are Log2 fold scale and significant differences are represented by blue dots. **b** Gene-set enrichment analysis for significantly upregulated genes in SCs. **c** Heatmap showing expression of *Fabp5* and various genes encoding metalloproteinases in SCs relative to other cell populations, namely diaphyseal MSCs (dpMSCs1 and dpMSCs2), metaphyseal MSCs (mpMSCs), osteoblasts (OBs), and proliferating bone mesenchymal stromal cells (p-BMSCs). **d, e** Maximum intensity projection of 3-week-old wild-type femur showing high MMP9 staining in SCs (arrowheads) (**d**). MMP13 (red) expression in MMP9+ (green) CD68- cells (white arrowheads) but also in CD68+ (blue) OCs (orange arrowheads) (**e**). **f** Confocal image showing MMP9 (green) immunosignals around ACAN+ (cartilage-specific proteoglycan core protein aggrecan) (red) hypertrophic chondrocytes (HC, arrowheads). **g** *Acan-CreERT2*-labelled (GFP, green) chondrocyte fragments in FABP5+ (red) SCs (white arrowheads) but not in CD68+ OCs (orange arrowheads) in proximity of the growth plate. **h** High magnification confocal images showing strong LAMP1 (green) signal in FABP5+ SCs (red, white arrowhead) relative to vATPase+ (red, orange arrowheads) OCs. DAPI (blue). **i** Double immunogold labelling showing strong LAMP1 staining (15 nm gold, yellow arrowheads) in SCs relative to vATPase-labelled (10 nm gold, orange arrowheads) OCs. ER, endoplasmic reticulum; LYS, lysosome. **j** FABP5+ SCs (arrowheads) decline in adult and ageing femurs relative to juvenile metaphysis. GP, growth plate; EPI, ephiphyseal line. Quantitation is shown on the right ($n = 6$ biologically independent samples; data are presented as mean values ± SEM, *p*-values. Statistical analysis was performed using Tukey multiple comparison test (one-way Anova). Source data are provided in a Source data file. **k** Schematic summary showing MMP secretion by EC-associated SCs to degrade growth plate matrix, and phagocytosis of hypertrophic chondrocytes (HC) debris. Independent animals, **d–h** ($n = 5$–6) and **i** ($n = 3$).

while *Jag1* is also found in mpMSCs (Fig. 4h). Consistent with the important role of Notch signalling in the vasculature[26,27], transcripts for the receptors *Notch1* and *Notch4* are abundant in ECs, whereas *Notch3* but also *Notch1* and *Notch2* are expressed by SCs (Fig. 4h). Furthermore, staining of bone sections shows that expression of the *Hey1-GFP* Notch reporter strongly labels SCs in addition to mpMSCs (Fig. 4i and Supplementary Fig. 8f). *Hey1-GFP*+ FABP5+ SCs are located close to endothelial buds, the most distal protrusions of growing CD31$^{high}$ EMCN$^{high}$ vessels in direct proximity of the growth plate, and bud ECs show substantial Dll4 immunostaining (Fig. 4j and Supplementary Fig. 8f). Further arguing for a role of Notch signalling, stimulation of FACS-isolated *Hey1-GFP*+ metaphyseal cells (Supplementary Fig. 8g) with immobilized recombinant Dll4 protein in vitro leads to significant increases in Notch target gene expression but also *Fabp5* and *Mmp9* transcripts (Fig. 4k). In vivo, both Dll4+ bud ECs and *Hey1-GFP*+ SCs are reduced in adult mice and disappear during ageing (Supplementary Fig. 9a). Our previous work has shown that inactivation of *Dll4* in EC impairs both bone angiogenesis and osteogenesis[28] (Fig. 5a and Supplementary Fig. 9b). Indicating direct crosstalk between Dll4+ ECs and *Hey1-GFP*+ SCs, GFP reporter signal is lost and FABP5+ cells are strongly reduced in EC-specific *Dll4* mutants (*Dll4*$^{iΔEC}$) relative to Cre-negative littermate controls (Fig. 5b–d and Supplementary Fig. 9c). The reduction in FABP5+ SC number is accompanied by significantly reduced MMP9 expression in *Dll4*$^{iΔEC}$ mutants, whereas hypertrophic chondrocytes and unresorbed chondrocyte fragments in the metaphysis are increased (Fig. 5e, f and Supplementary Fig. 9d). In contrast, CD68+ osteoclasts are not lost in *Dll4*$^{iΔEC}$ mutant bone samples (Supplementary Fig. 9e). These findings indicate that septoclasts are controlled by Dll4-expressing ECs through Notch signalling (Fig. 5g).

**Septoclasts in fracture healing.** The repair of fractured long bone is a multistep procedure involving haematoma formation, angiogenesis, formation of a chondrocyte and matrix-rich soft callus, and its gradual conversion into new bone[2,29]. scRNA-seq analysis of non-haematopoietic stromal cells during the bone repair phase at post-fracture day 14 (PFD14) and age-matched, unfractured controls resulted in data for 45,321 cells (Fig. 6a and Supplementary Fig. 10a). Cluster-specific marker genes were used to characterize different cell types and establish a map defining stromal heterogeneity and the changes during fracture repair (Fig. 6b–e and Supplementary Fig. 10b, c). Chondrocytes and fibroblasts are strongly increased in fractured bone, while dpMSCs are decreased relative to other populations at PFD14. The latter might reflect that part of the marrow in the fractured femur is blocked by the insertion of a stabilizing pin. EC number

is comparable in PFD14 bone and control samples, whereas the number of proliferating cells is substantially increased in fracture bone stromal cells (Fig. 6d, e and Supplementary Fig. 10d, e). Consistent with the decline of SCs in adult and ageing mice, *Fabp5*+ *Mmp9*+ cells were scarce in the control mpMSC cluster but increased during fracture repair (Fig. 6f, g). This was confirmed by subclustering analysis of mpMSCs, which also showed upregulation of *Fabp5* and *Mmp9* expression in SCs of PFD14 samples (Fig. 6h, i and Supplementary Fig. 10f). *Mmp13* expression is strongly increased and expands to different cell populations in fractured bone relative to control (Supplementary Fig. 10g). Trajectory analysis of cells in the PFD14 scRNA-seq data is consistent with the conversion of dpMSCs into mpMSCs, which, in turn, give rise to osteoblast lineage cells, fibroblasts, chondrocytes, and SCs (Fig. 7a, b and Supplementary Fig. 10h).

Consistent with the scRNA-seq data, immunostaining of PFD14 bone confirms extensive accumulation of chondrocytes and PDGFRβ+ mesenchymal cells in the callus together with active blood vessel growth (Fig. 7c, d and Supplementary Fig. 11a). Common features shared by PFD14 samples and growing, 3-week-old femoral metaphysis include the bud-shaped morphology of CD31$^{high}$ EMCN$^{high}$ vessels invading the avascular, chondrocyte-rich regions of the callus, the association of distal vessels with PDGFRβ+ mesenchymal cells, and the abundance of Osterix+ osteoblast lineage cells around the growing vasculature (Fig. 7e and Supplementary Fig. 11b–d). Consistent with the increases seen in the scRNA-seq results, SCs are abundant in healing bone fractures and are associated with vessel buds in proximity of callus chondrocytes (Fig. 7f–h). The same region also shows high expression of MMP9 and MMP13 in fracture repair bone sections (Fig. 7h, i and Supplementary Fig. 11e). While CD68+ osteoclasts are seen in proximity of SCs and chondrocytes, these cells are found throughout the newly formed bone, are not concentrated at the chondro-vascular interface, and show comparably weaker MMP9 immunostaining (Fig. 7j). In addition, expression of the transcription factor Runx2 and the angiogenic growth factor VEGF-A are strongly increased in BMSCs and chondrocytes of the callus (Supplementary Fig. 11f, g). These data support that regenerating bone recapitulates major features of developmental osteogenesis including the re-emergence of FABP5+ SCs as well as budding angiogenesis and high abundance of vessel-associated Osterix+ cells.

## Discussion

Fractures typically result from trauma or bone-weakening diseases such as osteoporosis, which affects more than 10% of the population above the age of 50 years in the USA and Europe[30,31]. The degradation of cartilage in the fracture callus is essential for

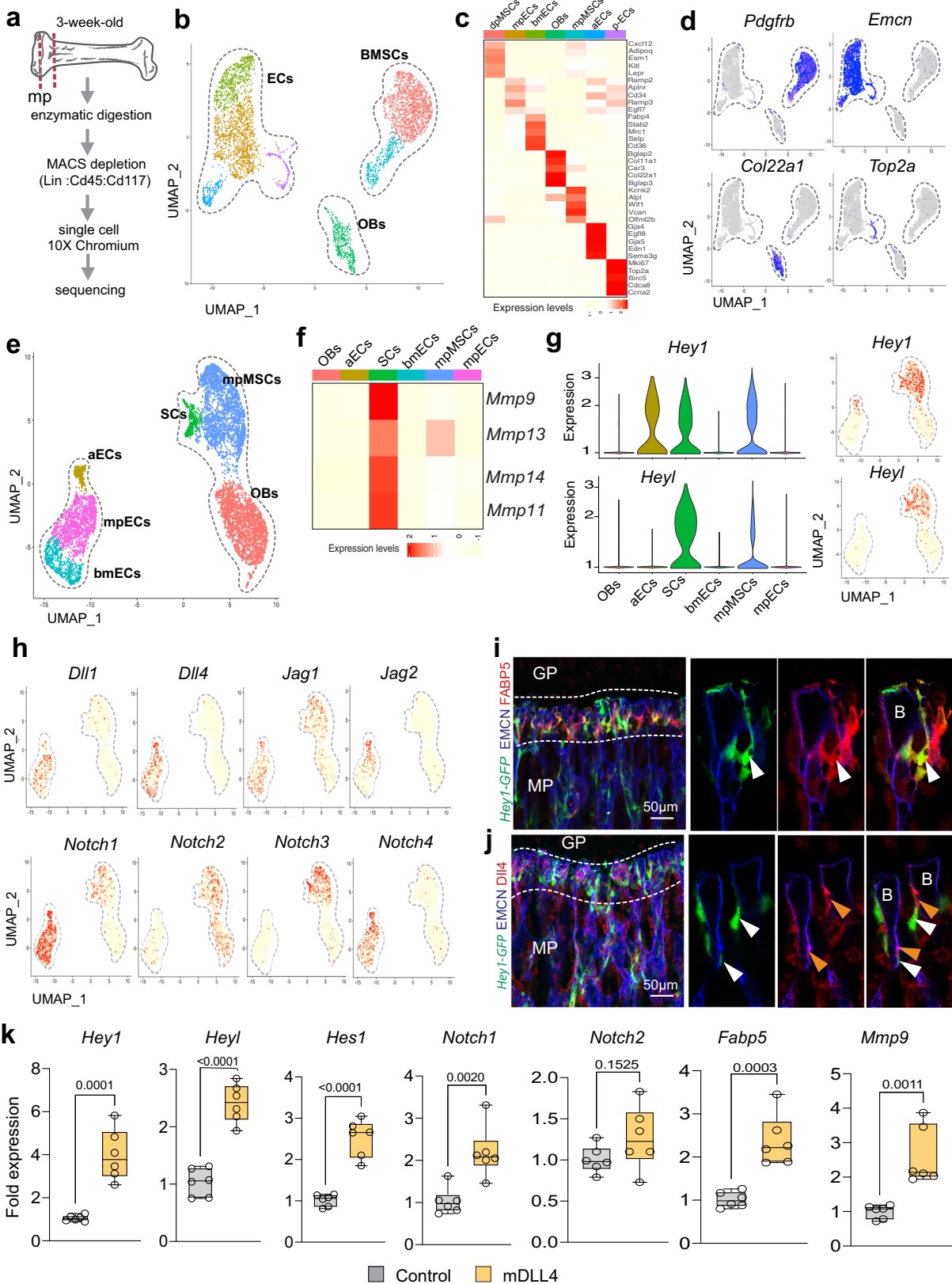

ossification, but the nature of cartilage-resorbing cells during regeneration as well as developmental osteogenesis has been a long-standing and controversial question. Based on the data presented above, we propose that cartilage resorption is mediated by mononuclear septoclasts, which express FABP5[11] but also, as we show, multiple secreted proteases and genes controlling

extracellular matrix organization. Unlike osteoclasts, SCs are not associated with calcified bone and show no expression of vAT-Pase, which is critical for local pH changes allowing the dissolution of calcium phosphate. This argues that septoclasts are unlikely to mediate the resorption of calcified bone. Furthermore, multiple lines of evidence, including scRNA-seq data,

**Fig. 4 Identification of Notch as mediator of EC–SC interactions. a** Preparation of non-haematopoietic bone stromal cells for scRNA-seq analysis. **b–d** UMAP plot showing colour-coded cell clusters in bone stromal cells (**b**). Top 5 marker genes shown in heatmap (**c**). Feature blots of cell type-specific markers: *Emcn*—Endothelial cells (ECs); *Col22a1*—osteoblastic cells (OBs); *Pdgfrb*—bone mesenchymal stromal cells (BMSCs); *Top2a*—proliferating cells (**d**). **e, f** Visualization of subclusters in merged scRNA-seq data from *Pdgfra-GFP*+ cells and non-haematopoietic stromal cells. Indicated are arterial (aECs), metaphyseal (mpECs) and bone marrow (bmECs) ECs in addition to SCs, OBs and metaphyseal MSCs (mpMSCs) (**e**). Heatmaps of *Mmp9*, *Mmp13*, *Mmp14*, and *Mmp11* expression in SCs relative to other cell populations (**f**). **g, h** Violin and UMAP plots showing high expression of Notch target genes *Hey1* and *Heyl* in SCs and lower levels in mpMSCs (**g**). Expression of Notch ligand transcripts and *Notch1* and *Notch4* by ECs. Multiple Notch receptor transcripts are also found in SCs and mpMSCs (**h**). **i, j** Confocal image showing *Hey1-GFP* (green) in FABP5+ (red) SCs (white arrowheads) near vessel buds (B) but also in mpMSCs in 3-week-old femur (**i**). DLL4 (red) marks bud ECs (orange arrowheads) next to *Hey1-GFP*+ perivascular SCs (white arrowheads) (**j**). Independent animals, **i, j** (*n* = 4). **k** RT-qPCR analysis shows that Notch target genes *Hey1*, *Heyl*, *Hes* and expression of *Fabp5* and *Mmp9* are significantly increased ex vivo by stimulation of SCs with immobilized recombinant mouse DLL4 (*n* = 6 control and mDll4 treated samples in three independent experiments. Data are presented as mean values ± SEM. Statistical analysis was performed using Mann–Whitney test (two-tailed). Source data are provided in a Source data file.

immunostaining for the receptor tyrosine kinases PDGFRα and PDGFRβ but also genetic lineage tracing, indicate that SCs are derived from BMSCs, whereas osteoclasts are polyploid cells emerging from the haematopoietic lineage. Signalling by PDGFRβ in response to secreted platelet-derived growth factor B (PDGF-B) triggers the activation of skeletal stem and progenitor cells for bone regeneration[32], which raises the interesting possibility that the formation of bone-forming cells and septoclasts are coupled during callus remodelling. Consistent with our findings, it was previously reported that septoclasts express PDGFRβ and NG2 in the developing murine tibia[14]. The same study has also proposed that septoclasts emerge from pericytes. However, scRNA-seq studies have so far failed to identify a distinct pericyte population in developing and adult bone[33–35], whereas the sum of our findings, including genetic fate tracking experiments, supports that septoclasts have a non-hematopoietic origin and are derived from mpMSCs. This conclusion is also consistent with our previous characterization of *Pdgfrb-CreERT2*-expressing cells in developing bone, which showed that the transgene initially labels mesenchymal stromal cells in the metaphysis, which later give rise to other mesenchymal cell populations including reticular cells in marrow and adipocytes[35].

Furthermore, we propose that septoclasts are controlled by Notch signalling through interactions with Dll4+ vessel buds at the chondro-osseous interface. Our previous work had revealed that EC-specific inactivation of *Dll4* or of *Rbpj*, which encodes an essential mediator of Notch-induced target gene expression, impairs developmental osteogenesis and leads to the accumulation of hypertrophic chondrocytes in the mutant growth plate[28]. Later work has attributed this defect to the high expression of proteinases by ECs relative to osteoclasts, arguing for role of the endothelium in cartilage resorption[10]. scRNA-seq data presented in our current study, however, indicate that the expression of bone protease genes is far higher in FABP5+ septoclasts than in ECs or BMSCs (Figs. 3c, 4f and Supplementary Fig. 6c). While this does obviously not rule out that ECs can actively contribute to cartilage resorption, we propose that ECs use the ligand Dll4 to activate Notch in septoclasts, which increases SC number and expression of *Mmp9* in these cells.

While it is appreciated that developmental and regenerative ossification share important features and involve active growth of blood vessels[2,36], we now show that the presence of vessel buds in direct proximity of callus chondrocytes resembles budding angiogenesis in the postnatal metaphysis. Moreover, we show that SCs are another common hallmark of both developmental and regenerative endochondral osteogenesis. Our study also provides a map of non-haematopoietic cells during fracture repair, which offers insight into the molecular profile of critical cell populations and interconversion processes. Our findings will be instrumental for the understanding of pathologies that impair bone growth and

regeneration but also for the development of therapeutic strategies aiming at improved fracture healing.

## Methods

**Animal models.** C57BL/6J male mice were used for all wild-type bone analysis. *Pdgfra-H2B-GFP* heterozygous knock-in reporter mice[18] were used for single cell RNA sequencing and labelling of BMSCs. *Pdgfrb(BAC)Cre-ERT2* (ref. [35]) or *Vav1-Cre*[37] transgenic mice were interbred with *Gt(Rosa26) ACTB-tdTomato-EGFP* reporter animals[19] for cell fate tracking. Chondrocytes were targeted using *Acan-CreERT2* (ref. [38]) mice. *Hey1-GFP (Tg(Hey1-EGFP)ID40Gsat)* reporter heterozygotes (http://www.gensat.org) were used as readout for Notch signalling. Mice carrying loxP-flanked alleles of Dll4 (*Dll4flox/flox*) mice[39] were bred to *Cdh5(PAC) Cre-ERT2* transgenic animals[40] to generate EC-specific and tamoxifen-inducible mutants (*Dll4iΔEC*). Cre-negative control littermates were subjected to the same tamoxifen administration regime as mutants. For inducible Cre-mediated recombination, pups received daily intraperitoneal injections (IP) of 50 μg of tamoxifen from postnatal day 1 (P1) to P3. Tamoxifen stocks were prepared as described previously[40].

For studies in embryos, *Pdgfra-H2B-GFP* males were mated with wild-type females, and vaginal plugs were checked every day in the morning. Embryos were harvested at gestational time points E15.5 to E17.5. For ageing study, wild-type and *Hey1-GFP* reporter males were analysed as juveniles (3–4 weeks), adults (11–12 weeks) and ageing mice (70–80 weeks), respectively.

For fracture healing experiments, 10-week-old female mice were anesthetized by intraperitoneal injection of a ketamine hydrochloride/xylazine mixture (100 mg/kg and 10 mg/kg body weight, respectively) before the left leg was fractured by three-point bending. Fractures were stabilized with an intramedullary nail (MouseScrew, 15.2 mm length custom made, RISystems AG, Landquart, Switzerland) as described previously[41]. Carprofen (4 mg/kg subcutaneously) was given as an analgesic for 3 days and further on at 24-h intervals as required. Mice were sacrificed by cervical dislocation at PFD14 and age-matched (12-week-old) wild-type females were used as control.

Mice were kept in individually ventilated cages (IVC), with constant access to food and water under a 12 h light and 12 h dark cycle regime. Air flow, temperature (21–22 °C) and humidity (55–60%) were controlled by an air management system. Animals were checked daily and maintained in specific pathogen-free (SPF) conditions. Sufficient nesting material and environmental enrichment was provided. All animal experiments were performed according to the institutional guidelines and laws, approved by local animal ethical committee and were conducted at the University of Münster and the Max Planck Institute for Molecular Biomedicine with necessary permissions (02.04.2015.A185, 84-02.04.2016.A160, 81-02.04.2017.A238, 81-02.04.2019.A164) granted by the Landesamt für Natur, Umwelt und Verbraucherschutz (LANUV) of North Rhine-Westphalia, Germany. Mouse lines and other unique biological materials described in this article are either available through stock centres, commercial suppliers or, upon reasonable request, the lead author.

**Sample processing and immunostaining.** Mice were sacrificed and long bones (femur and tibia) were harvested and fixed immediately in ice-cold 2% paraformaldehyde (PFA) for 6–8 h under gentle agitation. Bones were decalcified in 0.5 M EDTA for 16–24 h at 4 °C under gentle shaking agitation, which was followed by overnight incubation in cryopreservation solution (20% sucrose, 2% PVP) and embedding in bone embedding medium (8% gelatine, 20% sucrose, 2% PVP). Samples were stored overnight at −80 °C. 60–100 μm-thick cryosections were prepared for immunofluorescence staining[42].

After dissection, embryos were fixed in 4% paraformaldehyde overnight, then washed in PBS and transferred to cryopreservation solution for 6 h. Later, embryos were embedded in embedding medium, and stored at −80 °C.

Bone immunostaining performed as described previously[40,42]. Bone sections were washed in ice-cold PBS and permeabilized with ice-cold 0.3% Triton-X-100 in

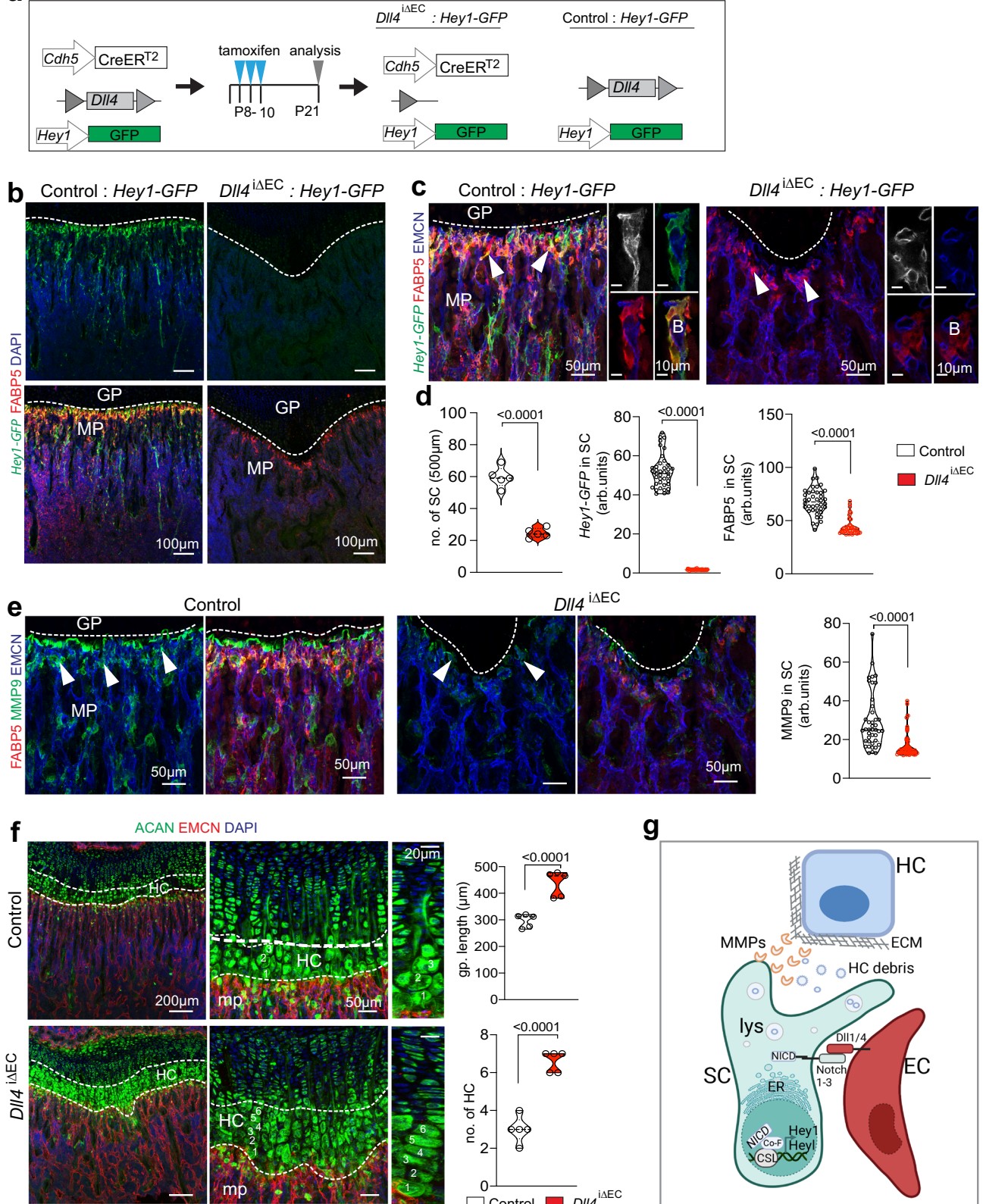

PBS for 10 mins at room temperature (RT). Samples were incubated in blocking solution (5% heat-inactivated donkey serum in 0.3% Triton-X-100) for 30 min at RT. Primary antibodies (rat monoclonal anti-Endomucin (V.7C7) (Santa Cruz, Cat# sc-65495, 1:100 dilution), goat polyclonal anti-CD31 (R&D, Cat# AF3628, 1:100 dilution), rabbit-polyclonal anti-Fabp5 (Lifespan Bioscience, Cat#C312991, 1:100 dilution), goat anti-Pdgfrb (R&D, Cat# AF1042, 1:100 dilution), rat monoclonal anti-CD68 (Abcam, Cat# ab53444, 1:200 dilution), rabbit-polyclonal

anti-Osterix (Abcam, Cat#ab22552, 1:300 dilution), rabbit polyclonal anti-NG2 (Millipore, Cat#AB5320, 1:100 dilution), rabbit monoclonal anti-CD146 (Abcam, Cat#ab75769, 1:100 dilution), Chicken polyclonal anti-GFP (Abcam, Cat#ab13970, 1:200 dilution), goat polyclonal anti-Mmp9 (R&D, Cat#AF909, 1:200 dilution), rabbit polyclonal anti-Mmp13 (Proteintech, Cat#18165-1-AP, 1:100 dilution), rabbit polyclonal anti-Mmp14 (Invitrogen, Cat#PA5-13183, 1:100 dilution), rabbit polyclonal anti-Acan (Milipore, Cat#AB1031, 1:100 dilution), rat monoclonal anti-

**Fig. 5 Endothelial Notch ligand Dll4 controls SC functions. a** Experimental scheme showing tamoxifen-inducible *Dll4* inactivation in ECs with *Cdh5-CreERT2* transgenic mice in the *Hey1-GFP* reporter background. **b** Representative confocal images showing strongly reduced *Hey1-GFP* expression after *Dll4* inactivation in ECs (*Dll4*$^{i\Delta EC}$) compared to *CreERT2*-negative control. Growth plate (GP, dashed line), metaphysis (MP) and distal vessel buds (B) are indicated. SCs, FABP5 (red); nuclei, DAPI (blue). Independent animals, **a**, **b** (*n* = 5). **c**, **d** Septoclast number and *expression of Hey1-GFP* and FABP5 (red) are reduced in *Dll4* $^{i\Delta EC}$ mutant bone metaphysis compared to control (arrowheads) (**c**). Quantification of data (arbitrary units) (**d**). (no. of SC *n* = 5 control and mutant bones; expression in SCs *n* = 4 bones × 10 cells; data are presented as mean values ± SEM. Statistical analysis performed using Mann–Whitney test (two-tailed). **e** Confocal images showing reduction of total and SC-associated (green, arrowheads) MMP9 immunostaining in *Dll4*$^{i\Delta EC}$ mutants relative to control. Graph shows quantification of MMP9 expression in SCs (arbitrary units) *n* = 4 bones × 10 SCs; data are presented as mean values ± SEM. Statistical analysis performed by Mann–Whitney test (two-tailed). **f** Confocal images showing accumulation of hypertrophic chondrocytes (HC) in *Dll4*$^{i\Delta EC}$ growth plate (GP). EMCN, ECs (red); DAPI, nuclei (blue). GP length and HC number are increased in mutants (*n* = 5 control and mutant bone; data are presented as mean values ± SEM, Statistical analysis performed using Mann–Whitney test (Two-tailed). **g** Proposed model of Dll4-controlled interactions between bud ECs and SCs in the regulation of MMP9 expression and resorption of hypertrophic chondrocytes. d-f, source data are provided in a Source data file.

Lamp1 (BD Pharmingen, Cat#553792, 1:100 dilution), rabbit anti-vATPaseB1/B2 (Abcam, Cat#200839, 1:100 dilution), rabbit polyclonal anti-Ki67 (Abcam, Cat# ab15580, 1:100 dilution), goat anti-Dll4 (R&D, Cat#AF1389, 1:50 dilution), rabbit monoclonal anti-Runx2 (Abcam, Cat#ab192256, 1:200 dilution), rabbit monoclonal anti-VEGF-A (EP1176Y) (Abcam, Cat#ab52917, 1:200 dilution) were diluted in 5% donkey serum mixed PBS and incubated overnight at 4 °C. Next, slides were washed 3–5 times in PBS in 5–10 min intervals. Species-specific Alexa Fluor secondary antibodies Alexa Fluor 488 (Thermo Fischer Scientific, Cat#A21208), Alexa Fluor 546 (Thermo Fischer Scientific, Cat#A11056), Alexa Fluor 594 (Thermo Fischer Scientific, Cat#A21209), Alexa Fluor 647 (Thermo Fischer Scientific, Cat#A31573 or Cat#A21447) diluted 1:100 in PBS were added and incubated for 3 h at RT.

**Safranin O staining**. Long bones from control and *Dll4*$^{i\Delta EC}$ mutants were harvested, fixed and decalcified as described above. Bone was dehydrated and embedded in paraffin wax by stranded histology methods. Bone paraffin block cut at 5 microns, slides were deparaffinize, and hydrate. Next, stain with 0.1% Safranin O for 8 min, washed with water. Next, slides were dehydrated and dried, mounted with non-aqueous mounting medium (Entellan, Sigma).

**Electron microscopy and immunogold labelling**. Femurs were removed from 3-week-old wild-type mice and directly cut into half in fixative. For ultrastructural analysis, the fixative comprised 2% glutaraldehyde, 2% paraformaldehyde, 20 mM CaCl$_2$, 20 mM MgCl$_2$ in 0,1 M cacodylate buffer, pH 7,4. For immunogold labelling, the sample was mildly fixed in 2% paraformaldehyde, 0,2% glutaraldehyde, 20 mM CaCl$_2$, 20 mM MgCl$_2$ in 0,1 M PHEM-buffer, pH 6,9 to preserve antigenicity.

The stronger fixed material was post-fixed in 1% osmium tetroxide containing 1.5% potassium ferrocyanide, dehydrated, including uranyl-en-bloc staining and embedded stepwise in epon. Ultramicrotomy was performed until reaching the area of interest, where 60 nm ultrathin sections were collected on formvar-coated 1 slot copper grids. Sections were further stained with lead citrate and finally analysed with a transmission electron microscope (Tecnai-12-biotwin, Thermo Fisher Scientific).

Mildly fixed samples were processed for cryo-immunogold labelling according to the Tokuyasu method[43]. For better orientation, bone was first sectioned with a vibratome and the sections were embedded in a layer of 10% gelatin. Samples were infiltrated with 2.3 M sucrose, put on pins and frozen in liquid nitrogen. From the area of interest, 50 nm ultrathin sections were cut at −110 °C. Sections were picked up with a mixture of sucrose/methylcellulose and thawed on formvar coated copper grids (200mesh, hexagonal). Double-immunogold labelling was accomplished using antibodies against Lamp1 (BD Biosciences, Cat#553792) and vATPase (Abcam, Cat#ab200839), and detected by 15 nm and 10 nm protein A gold, respectively (CMC, Utrecht, Netherlands). Representative images were taken with a 2K CCD camera (Veleta, EMSIS, Münster, Germany).

**Bone stromal cells preparation for scRNA-seq**. To enrich septoclasts, we used 3-week-old *Pdgfra-H2B-GFP* heterozygous males for single cell RNA sequencing analysis. Femur and tibia were harvested, cleaned from attached surrounding tissue, the metaphysis region was dissected and collected in digestion enzyme solution (Collagenase type I and IV, 2 mg/ml). Next, bones were cut into small pieces and crushed using mortar and pestle. Samples were digested for 20 min at 37 °C under gentle agitation. Digested samples were transferred to 70 µm strainers in 50 ml tubes to obtain a single–cell suspension, which was resuspended in blocking solution (1% BSA, 1 mM EDTA in PBS without Ca$^{2+}$/Mg$^{2+}$), centrifuged at 300 × *g* for 5 mins, washed 2–3 times with ice-cold blocking solution, and filtered through 50 µm strainers. Pellets were resuspended in respective volume of blocking solution. GFP+ cells were sorted from the single cell suspension with a FACS Aria II sorter (BD Bioscience), collected in blocking buffer and used for scRNA-seq.

For scRNA-seq of non-haematopoietic bone stromal cells, 3-week-old C57BL/6J male metaphysis segments were spatially dissected and single cell suspension was prepared as described above. Single cell suspensions were subjected to lineage depletion using lineage cell depletion kit (MACS, cat#130-090-858) following the manufacturer's instructions. Next, lineage negative cells were depleted by CD45 and CD117 using microbeads (MACS, cat#130-052-301 and cat#130-091-224) from lineage negative (lin-) bone cells to enrich bone stromal cells. The remaining cells were resuspended to final concentration of 10$^6$ cells/ml in 0.05% BSA in PBS, examined by microscopy and used for scRNA-seq.

For fracture bone scRNA-seq experiments, PFD14 and control femurs were harvested, cleaned from surrounding tissue and the nail insert was removed from fracture samples. Control and PFD14 bone single cell suspensions and non-haematopoietic stromal cells were prepared for scRNA-seq as described above.

Single cell suspensions were subjected to droplet-based scRNA-seq. Single cells were encapsulated into emulsion droplets using Chromium controller (10X Genomics). scRNA seq libraries were prepared using the Chromium single cells 3′ regent kit (V3) (10X Genomics, cat#PN-10000075) according to the manufacturer's protocol. scRNAseq libraries were evaluated and quantified by Agilent Bioanalyzer using High sensitivity DNA kit (cat#5067-4626) and Qubit (ThermoFisher Scientific, Cat# Q32851). Individual libraries were diluted to 4 nM and pooled for sequencing. Pooled libraries were sequenced by using High Output kit (150 cycle) (Illumina cat#TG-160-2002) with a NextSeq500 sequencer (Illumina).

**Fluorescence-activated cell sorting (FACS)**. Single cell suspensions were prepared from 3-week-old *Pdgfra-H2B-GFP* reporter mice femur and tibia bone as described above. Co-culture Chondrocytes (*AcanCre-tdTomato*) and septoclasts single cell were prepared. Next, Cell sorting was performed on a FACS Aria II cell sorter (BD Biosciences). Dead cells and debris were excluded by FSC, SSC and DAPI positive signal. Sorted GFP positive cells were collected in blocking buffer for scRNA-seq.

**Isolation, culture and co-culture of septoclasts and chondrocytes**. For septoclasts, Femur and tibia were harvested from 3-week-old *Hey1-GFP* reporter mice. Metaphyseal (mp) region was separated and removed marrow cells by rinsing with ice-cold PBS. Single cells suspensions were prepared from mp and sorted GFP$^{high}$ cells by FACS. Sorted GFP cells cultured and expanded in MesenCult expansion medium (StemCell technology) at 37 °C in 5% CO$_2$ in a humidified atmosphere. 2–3 passages were used for all experiments.

Chondrocytes were isolated from wild-type and *AcanCre-dTtomato* femur and tibia of postnatal day 6 (p6) C57BL/6J pups growth plate. Growth plates were cut into small pieces and digested with Collagenase type I and II (2 mg/ml) for 45 min at 37 °C. Single cell suspensions were prepared as described above. Chondrocytes were cultured in DMEM medium (ThermoFisher Scientific) supplement with 10% FBS+ 2 mM L-Glutamine and 5 ml penicillin-streptomycin for 500 ml medium at 37 °C in 5% CO$_2$ in a humidified atmosphere.

For co-culture of septoclasts and chondrocytes, cells were seeded in 4 well insert micro-dish (ibidi). Inserts were removed at day 3 and cells were cultured 5 more days allowing direct cell-cell interactions. Next, cells were washed with PBS and fixed for immunofluorescence staining.

For experiments with culture-treated and calcium phosphate-treated Osteo assay surface 24 well plates (Corning, CLS3987), septoclasts were kept for 5 days at 37 °C in 5% CO$_2$ in a humidified atmosphere. Next, cells were washed with PBS and fixed with 4% PFA for 30 min at room temperature and followed by Von Kossa staining. Fixed septoclasts were washed with PBS and water, cells were stained with 5% silver nitrate for 30 min under UV light. To stop the silver nitrate development, cells were washed with water two times and treated with a 5% sodium thiosulfate solution. Following another water wash, samples were air dried and resorption areas were visualized using light microscopy.

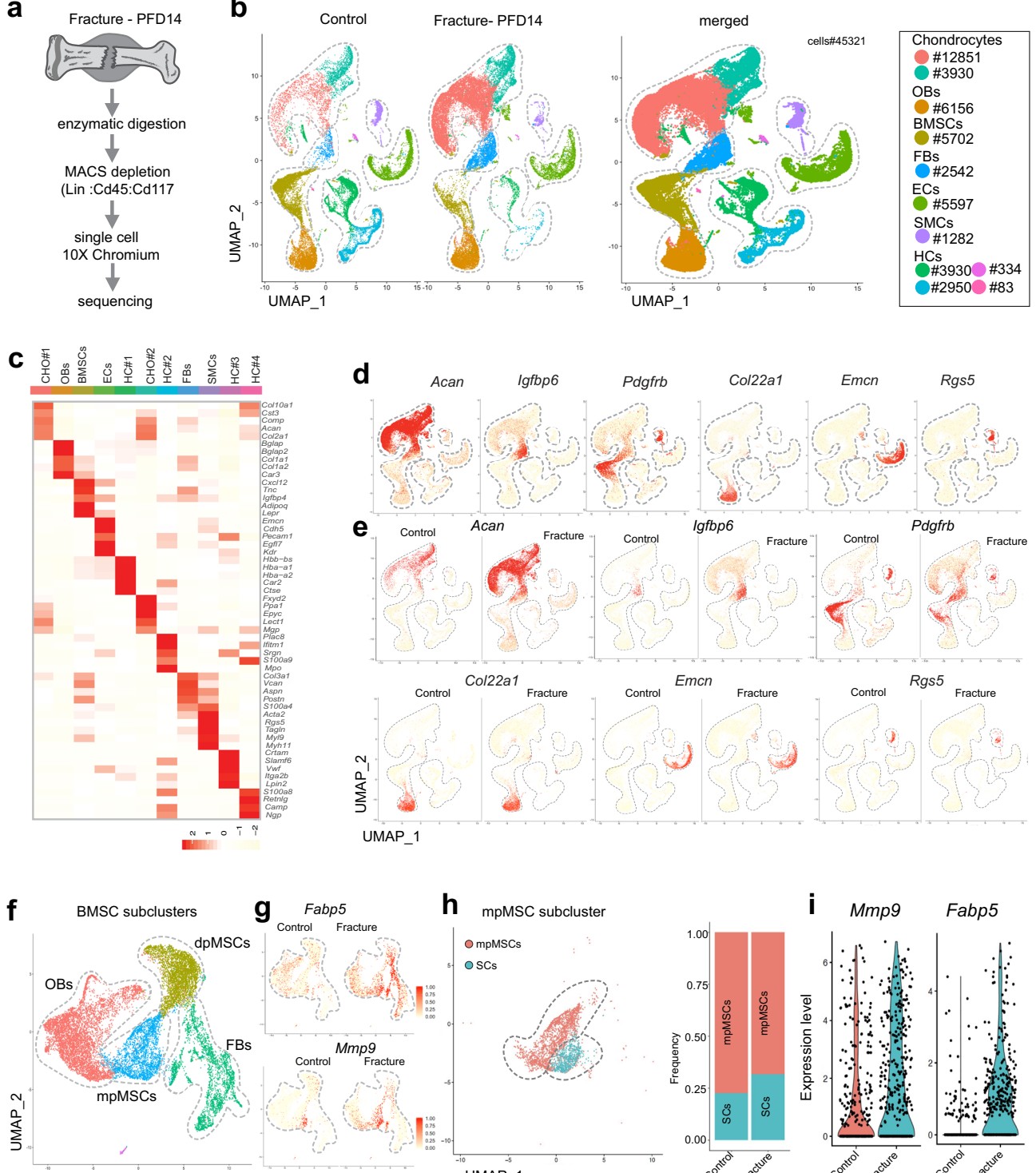

**Fig. 6 Single cell RNA-sequencing analysis of healing bone fractures. a** Preparation of non-haematopoietic stromal cells from PFD14 and control bone for scRNA-seq analysis. **b** UMAP plots showing colour-coded cell clusters from control and PFD14 bone. Cell types and numbers per cluster are displayed on the right. **c–e** Heatmap showing the top 5 marker genes for each cluster (**c**). Feature blots showing markers for non-haematopoietic bone cells: *Acan*—chondrocytes (CHO); *Igfbp6*—fibroblasts (FB); *Pdgfrb*—bone mesenchymal stromal cells (BMSCs); *Col22a1*—osteoblast lineage cells (OBs); *Emcn*—ECs; *Rgs5*—Smooth muscle cells (SMCs) (**d**). Comparison of marker gene expression in PFD14 and control samples (**e**). Haematopoietic cells (HCs-1,2,3,4). **f, g** UMAP plots showing colour-coded BMSC subclusters from control and PFD14 bone (**f**). *Fabp5* and *Mmp9* expressing cells are increased in PFD14 samples (**g**). **h, i** UMAP plot showing mpMSCs and SCs in the control and PFD14 mpMSC subcluster (**h**). Fracture-derived SCs show higher expression of *Mmp9* and *Fabp5* (**i**).

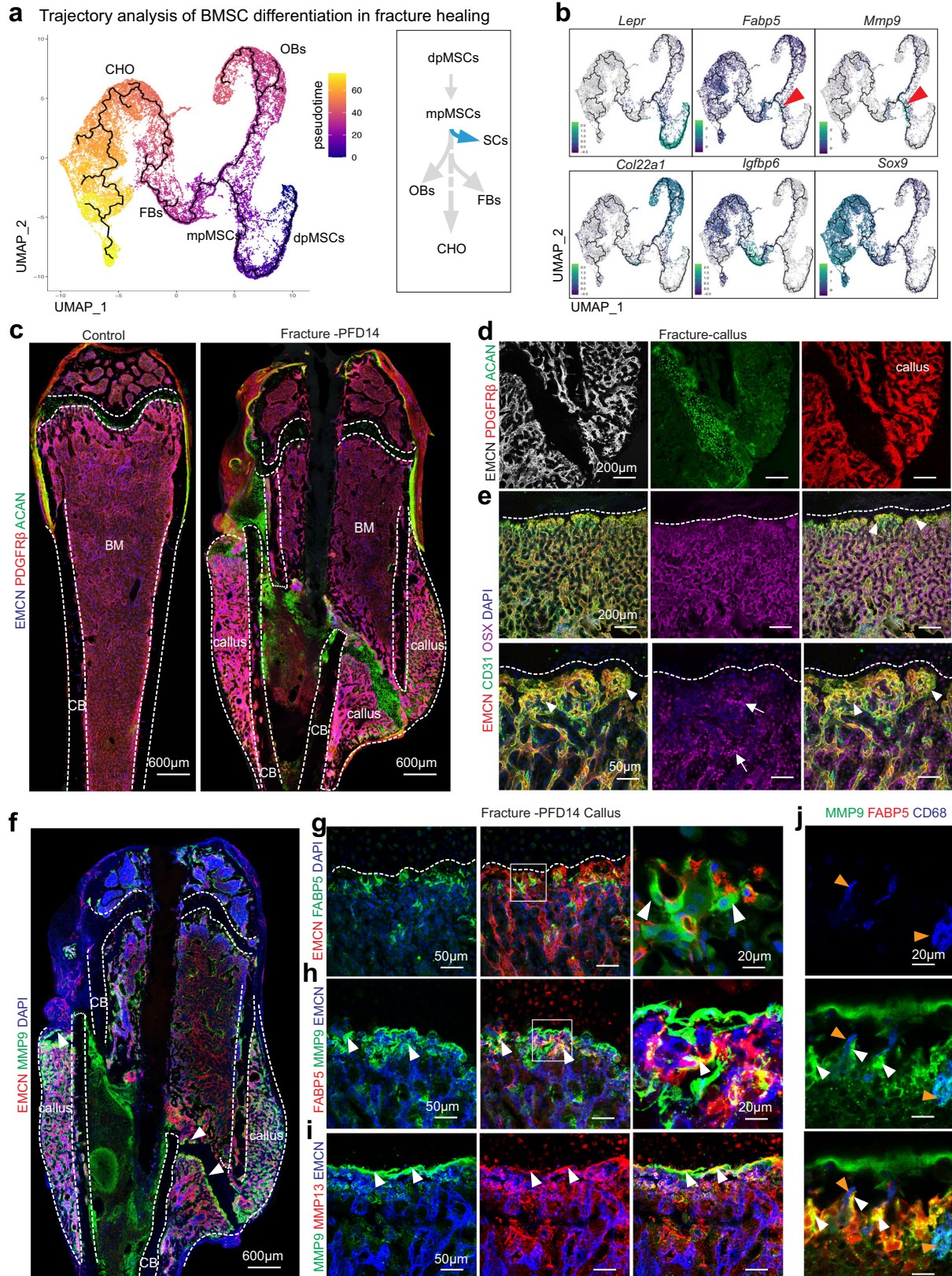

**Activating Notch signalling in septoclasts**. For the stimulation of Notch signalling, 6-well plates were incubated with anti-His (zymed) antibodies for 45 min at 37 °C. Plates were washed with PBS and blocked with 10% FCS in DMEM for 45 min at 37 °C. Next, Plates were washed with PBS and recombinant Dll4 (mouse Dll4-His, R&D; #1389-D4) was diluted in PBS to a concentration of 2ug/ml and 700ul of ligand mix were added per wells to a 6-well plate and incubated for 2 h at

37 °C and washed with PBS. Septoclasts were plated for 8 h at 37 °C in 5% Co2 in a humidified atmosphere.

**Quantitative PCR (qPCR)**. For gene expression analysis, total RNA was isolated from septoclast by using RNA Plus mini kit (Qiagen, Cat#74134) according to the

**Fig. 7 Septoclasts in fracture healing. a, b**. Monocle trajectory analysis of BMSC lineage cell differentiation path during fracture repair (**a**). Expression of marker genes is displayed in trajectory. Red arrowhead indicates *Mmp9+* and *Fabp5+* cells in proximity of mpMSC cluster (**b**). **c–e** Tile scan confocal images of PFD14 and age-matched control femur showing ECs (EMCN, blue), BMSCs (PDGFRβ, red) and chondrocytes (ACAN, green) (**c**). High magnification images showing avascular regions of callus containing ACAN+ chondrocytes and PDGFRβ+ BMSCs associated with vessels (**d**). Low and high magnification images showing CD31[hi] (green) and EMCN[hi] (red) vessels and endothelial buds (arrowheads) in proximity of callus chondrocytes. OSX+ (purple) cells (arrows) are abundant around vessels (**e**). **f–j** Tile scan confocal image of PFD14 femur showing MMP9 (green) at the vessel front in the callus (arrowheads) and, at lower level, in the metaphysis near growth plate. Dashed lines mark cortical bone (CB), bone marrow (BM), callus area, and growth plate (**f**). Representative confocal image of PFD14 callus with FABP5+ (green) SCs associated with distal EMCN+ (red) vessels (arrowheads) (**g**). High magnifications images show MMP9 and MMP13 immunosignals in the region containing SCs (arrowheads) (**h, i**). Weaker MMP9 signals relative to SCs (white arrowheads) mark CD68+ (blue) OCs (orange arrowheads) in the PFD14 callus (**j**). **c–i** (control = 5 and fracture = 4) independent biological samples.

manufacturer's instructions. RNA was reverse-transcribed using the iScript cDNA synthesis kit (Bio-Rad, Cat#1708890). Quantitative PCR was carried out using gene specific Taqman probes (Thermofisher), Hey1 (Mm00468865_m1), Heyl (Mm00516558_m1), Hes1 (Mm01342805_m1), Notch1 (Mm00435245_m1), Notch2 (Mm00803077_m1), Mmp9 (Mm00442991_m1), Fabp5 (Mm00783731_s1), Acan (Mm00545794_m1), Mmp14 (Mm00485054) and Mmp13 (Mm00439491). Gene expression was normalized to eukaryotic 18 s rRNA (Thermofisher, 4319413E). Quantitative PCR were performed in C1000 Touch Thermal cycler (BIORAD).

**Single cell RNA sequencing data analysis**

*Read data pre-processing and read mapping.* Sequencing results were demultiplexed and converted to FASTQ format using Illumina's bcl2fastq software. Raw reads were processed using fastp[44] (version 0.20) excluding reads with an average quality score of less than 20, trimming the ends of reads base by base with a quality score less than 20 and requiring all reads to be their minimum expected length, based on the number of cycles during sequencing for the first read, containing the cellular barcode and UMI.

Pre-processed read data were aligned to the mouse reference genome (mm10, Gencode M23 with H2B-GFP added) with STAR[45] (version 2.7.3a) through its single-cell functionality STARsolo, using parameters to mirror unfiltered 10x Genomics CellRanger 3 output for sample demultiplexing, barcode processing and transcript counting. The official 10× Genomics whitelist for the respective Chromium version was used.

*Count matrix pre-processing.* Datasets were analysed with default settings of the respective tools, if not specified otherwise. Preprocessing of the feature-count-matrix output by STARsolo was performed in R[46] (version 3.6.0) using the *scran*[47] (version 1.14.6), *scater*[48] (version 1.14.6), *SingleCellExperiment*[49] (version 1.8) and *ggplot2*[50] (version 3.3.2) packages. All visualisation was done using ggplot2 functions. Raw count matrices were loaded into SingleCellExperiment objects for storage and manipulation. The *EmptyDrops* function from the DropletUtils package[51] was used to pre-filter the feature-count-matrix. Per cell and per feature metrics were calculated with the appropriate functions of the scater package and used to exclude low-quality cells. Cells with less than 1500 (mpMSCs) or 1000 (*Pdgfra-GFP*) features, respectively, as well as more than 15% transcripts of mitochondrial origin and 30% of ribosomal origin were filtered out. By these strict thresholds mostly remaining cells of the haematopoietic lineage and low-quality cells were removed. Repeating the analysis without these cut-offs revealed no additional desired cell types (not shown). Only features present in more than 10 cells were retained. A total of 4,386 cells for the mpMSCs datasets and 11,242 for the Pdgfra-GFP dataset were used for further analysis. The mean numbers of detected genes per cell ranged from 2120 in the *Pdgfra-GFP* dataset to 2676 in the second replicate of the mpMSCs dataset.

*Clustering and visualisation.* Default settings were used, if not specified otherwise. Data normalisation was performed using the *normalize* function provided by the scran package. The identification of highly variable genes, Cell cycle scoring, UMAP[52] dimensionality reduction, Louvain[53] clustering with multilevel refinement and the identification of marker genes were performed using Seurat[54] (version 3.1.5). Always the top 5000 highly variable features were reported and used for principal component analysis (PCA)[55,56]. All clustering steps were done using 500 starts and 100 iterations per start. Marker genes were calculated with Seurats *FindAllMarkers* function, using its implementation of the non-parametric Wilcoxon Rank sum test. Cell identities were manually annotated based on known marker genes[33,57].

*Pdgfra-GFP datasets.* Variable genes were determined and then used for PCA. Cell cycle phases were classified and all genes were scaled and centred. For this scaling step only, the sum of mitochondrial reads per cell and the difference between the G2M and S phase scores as calculated earlier were regressed out. The first 30

principal components (PCs) were chosen and used for UMAP visualisation and clustering with multilevel refinement (resolution set to 0.6). Based on manually determined cell identities, the dataset was subset. Scaling, PCA, UMAP visualisation and Louvain clustering were rerun on the subset *Pdgfra-GFP* dataset using the first 15 PCs with a resolution of 0.3. Filtering this way had to be repeated once more, with the same settings, to remove a small proliferative cell population that was not properly captured in the clustering steps before.

*mpMSCs dataset.* The procedure was highly similar to the analysis of the *Pdgfra-GFP* dataset. To account for possible systematic technical effects between the replicates, the two pre-processed replicates were integrated using Seurat's alignment method for data integration[58,59] using the first 40 PCs. Briefly, this method uses canonical correlation analysis to learn the shared gene correlation structure across two datasets and then aligns them in lower-dimensional space thereby correcting for batch-effects. After integration, all genes were scaled and centred and the first 20 PCs were chosen for UMAP visualisation and Louvain clustering at a resolution of 0.4. Cell identities were determined as before and only cell identities of interest were retained. Scaling, PCA, UMAP visualisation and Louvain clustering were rerun, using the first 20 PCs and a clustering resolution of 0.4.

*Merged datasets.* The pre-processed mpMSCs and the final subset *Pdgfra-GFP* datasets were integrated using Seurat's integration method as before. Scaling, PCA, UMAP visualisation and Louvain clustering were run on the merged dataset using the first 25 PCs and a clustering resolution of 0.4. In this fashion also the following final clustering step was performed. After removing all cell identities not of interest for further analysis, a dimensionality of 15 PC was used for UMAP visualisation and a resolution of 0.3 for Louvain clustering.

**Fracture datasets.** For initial quality control of the extracted gene-cell matrices, we filtered cells with parameters low.threshold = 500, high.threshold = 6000 for number of genes per cell (nGene), high. threshold = 25% for percentage of mitochondrial genes (percent.mito) and genes with parameter min.cell = 3. Filtered matrices were normalized by LogNormalize method with scale factor = 10,000. Variable genes were found with parameters of selection.method = vst and nfeatures = 2000, trimmed for the genes related to cell cycle (GO:0007049) and then used for data integration (IntegrateData), data scaling (ScaleData) and principal component analysis (RunPCA). Statistically significant principal components were determined by JackStraw method and the first 12 principal components were used for non-linear dimensional reduction (UMAP) and clustering analysis (FindNeighbors) with resolution = 0.1.

For subclustering analysis of BMSCs, corresponding clusters were isolated using subset function, split into samples and then re-analysed from the variable feature identification (FindVariableFeatures). The first 13 principal components were used for downstream analyses. Resolution = 0.08 was used for the clustering of major cell populations and Resolution = 0.5 was used to annotate the population of SCs separately.

*Trajectory analysis.* Monocle[60] (version 2.14) and Monocle3 (version 1.0.0) was used for pseudotime trajectory analysis. Using monocle, we imported all information from Seurat objects to Monocle CDS objects and then performed dimensionality reduction using its DDRTree method with parameters max_components=2 and norm_method = "vstExprs". The top 100 marker genes as determined earlier by log2 fold-change for each cluster were used as ordering genes for the trajectory. The resulting Trajectory was then plotted with default settings. For trajectory analysis of BMSCs using Monocle3, expression matrix, cell names and gene names were extracted from Seurat object and then converted to Monocle3 object (new_cell_data_set). We calculated size factors (estimate_size_factor), performed principal component analysis computing 10 principal component (preprocess_cds) and dimensionality reduction using UMAP (reduce_dimension). Trajectory of single cells was constructed using functions, learn_graph and order_cells with default parameters. Results were visualized by plot_cells function.

**Statistics and reproducibility**. Immunostained bone sections were imaged with a Leica SP8 confocal microscope. Images were analysed, quantified and processed using Volocity (Perkin Elmer), Adobe Photoshop and Illustrator software 2020.

Statistical analysis was performed using Graphpad Prism 9 software or the R statistical environment (http://r-project.org). All data are presented as mean values ± SEM unless indicated otherwise. All box plots show whiskers down to the minimum and up to maximum value, and median with error bar. Comparisons between two groups were performed with unpaired two-tailed Mann–Whitney test. Comparisons between more than two groups were made by Tukey multiple comparison test to determine statistical significance. $P < 0.05$ was considered significant unless stated otherwise. Sample numbers are indicated in figure legends and were chosen based on experience from previous experiments. Reproducibility was ensured by several independent experiments. No animals were excluded from analysis.

**Reporting summary**. Further information on research design is available in the Nature Research Reporting Summary linked to this article.

## Data availability

The *Pdgfra-GFP* and metaphyseal bone stromal cell scRNA-seq data generated in this study have been deposited in the gene expression omnibus (GEO) database under the accession number "GSE154076". Bone fracture scRNA-seq data is available under accession number "GSE154247". The mouse reference genome (mm10, Gencode M23, https://www.gencodegenes.org/mouse/release_M23.html) was used for mapping the reads in this study. All other relevant data supporting the key findings of this study are available within the article and its Supplementary Information files or from the corresponding author upon reasonable request. Source data are provided with this paper.

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

## Acknowledgements

We thank M. Stehling for cell sorting. K. Mildner for helping EM sample preparation and immunogold labelling. This work was supported by the Max Planck Society, the University of Münster, the European Research Council (AdG 786672 PROVEC), the DFG (CRC1366), and the Leducq Foundation (RHA). RS, MT and RHA were supported by the Cluster of Excellence 1003 (EXC 1003) Cells in Motion (CiM).

## Author contributions

K.K.S. and R.H.A. designed experiments and interpreted results. K.K.S. generated and characterized mouse mutant lines, and conducted all experiments including FACS, bone sectioning and staining, cell biology, confocal imaging, quantifications, and sc-RNA sequencing. P-G.M., H.-W.J. and K.K.S. analysed the scRNA-sequencing data. B.D. staining and FACS. D.Z. performed immune-gold labelling and electron microscopy. S.S for technical assistance. R.S., M.T. and G.B. performed all bone fracture experiments. K.K.S. and R.H.A. wrote the manuscript.

## Funding

## Competing interests

The authors declare no competing interests.
