## [Peer Review File · Nature Communications]

Reviewers' Comments:

Reviewer #1:

Remarks to the Author:

Sivaraj et al. report that FABP5-expressing septoclasts (SC) at the chondro-osseous border zone are responsible for cartilage matrix degradation and chondrocyte phagocytosis. The authors report that septoclasts have a mesenchymal origin and that interaction with endothelial cells through Notch signaling is important for the identity of septoclasts. The use of several mouse models and numerous scRNA-seq analyses provide novel data, but they are mainly descriptive and sufficient functional prove that septoclasts are the cells that resorb the cartilage matrix is lacking.

Major comments:

1. Typical for osteoclasts is that they secrete not only proteases to cleave proteins, but also produce protons within a sealing zone to dissolve the calcium-phosphate crystals deposited in the matrix. In the present study no information is provided whether septoclasts can also dissolve the calcium-phosphate crystals of the calcified cartilage matrix or that other cell types are still necessary for mineral removal.
2. The authors describe that FABP5 is mainly expressed by cells at the chondro-osseous border. However, in Fig. 1a FABP5 staining is also observed in the bone marrow compartment of the metaphysis and diaphysis. Image magnifications of these regions should be provided (separate DAPI-FABP5 channels). Which cell types in the bone marrow are expressing FABP5? These data suggest that FABP5+ cells are a heterogenous population and this heterogeneity will influence the interpretation of the scRNA-seq data. It is therefore crucial to thoroughly describe (and identify) the different FABP5 populations.
3. The authors describe PDGFR α / β to be expressed by SCs and show co-localization with FABP5. The percentage of FABP5+ cells that express PDGFR α should be quantified in order to show how good the expression-overlap is? This information is important for the interpretation of the scRNA-seq analysis, as part of the FABP5+ SCs population may be not included in the analysis as only PDGFR α -GFP cells were considered for analysis.
4. The authors propose that SCs and not osteoclasts (or other cells) are responsible for cartilage degradation during endochondral bone formation. However, the evidence for this remains rather descriptive and not all data are supporting this claim. Fig. 3d shows that MMP9-expressing cells most closely localized to the growth plate are not FABP5+ cells, indicating that another cell type may be responsible for matrix degradation. In addition, the authors claim that the lysosomal protein LAMP1 is detected in septoclasts, but not in osteoclasts. Nevertheless, lysosomes are considered an integral part of osteoclast functioning and Lamp1 has been detected in these cells (Ferron M., Genes & Development, 2013), questioning thus the technicality of the staining. Proof that lysosomes are not present in these osteoclasts should be provided. Furthermore, the presence of lysosomes is not sufficient to support the claim that septoclasts are resorbing the cartilage matrix, as lysosomes are present in several cell types simply for proteostasis. In addition, detection of GFP-positivity in FABP5+ cells (Fig. 3g) can be derived from GFP leakage out of the dying chondrocytes and does not proof that septoclasts phagocytose chondrocyte fragments. Therefore, functional proof should be provided to support the claim that septoclasts resorb cartilage matrix. A possible approach to support this claim is to perform in vitro co-culture of chondrocyte cultures/micromasses with sorted SCs or sorted osteoclasts to indeed show that SCs are capable to resorb cartilage.
5. The authors inactivate Dll4 specifically in ECs to study the importance of Notch signaling for SC control via Hey1-GFP reporter mice. However, scRNA-seq data in Fig. 4h show that Hey1 is also expressed in ECs and mpMSCs. Therefore, Dll4 deletion can also affect the other Hey1-expressing cell types. The authors should provide proof that the observed phenotype of increase in hypertrophic chondrocytes is solely due to impaired SCs function? In addition, it is known that MMP9 is also produced by ECs: is the observed decrease in MMP9 due to decreased SC or EC function?
6. In Fig. 5d, the authors quantify the number of SCs. In Fig. 1c, SCs were counted based on

FABP5 expression, but in Fig. 5d a difference is made between SC and FABP5 in SC. How are the SC identified if not on FABP5 expression and how are these quantifications performed?

7. The authors report that SC number and MMP9 expression are decreased upon Dll4 deletion. To prove that SCs are responsible for cartilage resorption during bone development and fracture healing, the authors should provide Safranin O or Collagen2 staining to investigate whether cartilage remnants are observed, which would indicate defective cartilage resorption in Dll4 mutants? This experiment would provide additional evidence that SCs are responsible for cartilage degradation.

Minor comments:

1. Fig. 1d. The identification of the different cell types by electron microscopy looks rather arbitrary as no information is provided what the basis was for identifying some cells as being septoclasts. A similar comment applies to fig 3i where it is not clear on which basis cells are considered to be SC or OC. In addition, the labeling of Lamp1 vs vATPase is not clear.

2. Ext. Fig. 2a. FABP5+ cells are described as being highly proliferative, but no quantification is performed and the images show only a few EdU -positive FABP5+ cells.

3. Fig. 2b-c: dpMSCs and mpMSCs are described as different cell populations based on respectively Esm1 and Postn expression. The authors should provide evidence that these markers are specific for stromal cells in either the diaphyseal or metaphyseal region respectively.

4. Fig. 2a and Ext Fig. 3h: since it is claimed that septoclasts are not derived from the hematopoietic lineage it is important to show images with localization of GFP and FABP5+ cells (red) without any other staining. Now the 3 stainings are showed together (GFP, FABP5 and CD68). Especially, in the bottom panels, large green cells (clusters) are observed almost at the same location as FABP5+ cells.

5. Fig. 3b and Extended Fig. 4a-b: information should be provided whether GSEA data show the top-ranking upregulated pathways or whether a selection of pathways-of-interest was made for the visualization? Please include this information in the figure and/or text.

6. Figure 6: scRNA-seq is performed on control and fractured bone, but no information is provided which region of the fractured and control bone was used. This information is important as chondrocytes were also detected in the control condition.

Reviewer #2:

Remarks to the Author:

This paper addresses with the most modern techniques the cellular origin of the septoclasts, a cell type popularized mainly by the Amizuka group and that is supposed to resorb cartilage. Resorption of cartilage is critical during embryonic development to initiate ossification in bone forming through endochondral ossification. The main contribution of this paper which is quite important for the field is to show that septoclasts are of mesenchymal origin. The demonstration here is very convincing. There are however several concerns that need to be addressed. Some of them are experimental others in the interpretation of the results.

- Most experiments presented are done after birth and often in 3 week-old mice. Yet, as the authors present in the introduction, resorption of cartilage during development is absolutely necessary for endochondral ossification to proceed. Therefore what is missing from the paper is the demonstration that FAB5+ septoclasts can be seen later during embryonic development when cartilage is resorbed and ossification actually starts. Therefore, data that are presented in Figure 1a needs also to be shown in E17.5 or E18.5 embryos.

- FAB5+ septoclasts are present in the cortical bone (Figure1a), a fact that is not mentioned by the authors. What may be the function of these cells there? do they express the same genes that close to the growth plate?

- Another question raised by the data presented in Figure 1b is why do the number of septoclasts is lower at P1 than at p14? Also what happens between P14 and P28 Figure3j when their number begins to decline?
- An important question that needs to be addressed is to know if septoclasts are present or absent in bones that form through intramembranous ossification. This should not be difficult to achieve.
- The authors show clearly that septoclasts express at high levels MMPs (Figure3) and this is unquestionable, however there is only a correlation between the expression of the MMPs and Lamp1 in FAB5+septoclasts and the fact they may be resorbing cartilage. The demonstration has not been made in this paper. This is fine but the title of the paper, and throughout the results section should be rewritten in a more cautious manner.
- There is a quantification of the number of septoclasts in Figure 2 but not in Figure 4, could the authors provide this quantification?
- In the last 2 figures the authors address whether septoclasts are present during fracture healing. They show clearly that it is the case in a bone that develop through endochondral ossification. A control of specificity is to show if it is the case or not in fracture healing in bones developing through intramembranous ossification.
- What the authors show that is new in Figure 7 is that septoclasts re-emerge during fracture repair. They don't show "that regenerating bone recapitulates major features of developmental osteogenesis"because that is already well established. The authors need to also qualify this statement and stay closer to the data.

Reviewer #3:

Remarks to the Author:

General comments:

In this study, the authors identify septoclasts, defined by their expression of FABP5, to be responsible for chondrocyte phagocytosis and matrix degradation in the chondro-osseous border of the growth plate. They further demonstrate that these septoclasts are of mesenchymal origin – unlike osteoclasts – by lineage tracing. Septoclast specification is claimed to be mediated by endothelial cell via notch signalling. While septoclasts disappear with time after development, they re-emerge for bone repair.

The main novelty of the presented work is the identification that such septoclast population derive from stromal/mesenchymal progenitors (in contrast to hematopoietic-derived osteoclasts). This is investigated by extensive immunofluorescent stainings, lineage tracing and combined scRNA sequencing. However, the authors do not convincingly distinguish the identified cell population from perivascular cells and, while the mesenchymal origin is demonstrated for some cells, hematopoietic origin of at least some septoclasts cannot be entirely excluded.

Major issues:

1. The here shown septoclasts display several similarities with perivascular cells, such as the location, the mesenchymal origin, PDGFRB expression and their re-activation upon injury. Indeed, former studies suggest septoclasts to originate from pericytes, due to such co-expression (Bando 2018 Histochemistry and Cell Biology). Here the authors, however, conclude, that septoclasts are not pericytes because "...unlike pericytes, they are not embedded in the subendothelial basement membrane" p.4. line 86f. In order to distinguish clearly septoclasts from pericytes, better characterisation is required, such as staining for the expression of pericyte markers (e.g. NG2, CD146) and for nestin.

2. While the mesenchymal origin of FABP5+ cells is demonstrated by deriving from PDGFRB labelled cells, it looks like some FABP5+ cells also express CD68/Vav. This could indicate that at least a fraction of these cells may be of hematopoietic origin ("yellow" cells in Fig. 1C and Fig 2h).

This should be clarified and discussed.

3. The here presented data related to notch signalling mediated septoclast activation are rather descriptive and lack the functional prove, that septoclast activation is regulated by endothelial cells via *Dll4* activating the notch pathway. Could their "disappearance" in *Dll4* mutants also result from physical removal of endothelial cells and/or as a consequence of generally reduced vascularisation? To demonstrate direct involvement of notch signalling on septoclast activation, in vitro studies with isolated septoclasts should be performed, in which activation and inhibition of the notch signalling pathway can be demonstrated in a cleaner experimental setting. Additional co-culture experiments with endothelial cells expressing or not *Dll* could further imply a possible direct interaction between these cell types.

The here provided data so far do not support the statement in the discussion (p11, line 268f) that "Septoclasts are also controlled by Notch signalling through interactions with *Dll4*+ vessel buds at the chondro-osseous interface in the fracture callus"

Additional comments:

4. The aim of the study is rather generic: "...systematically investigate the processes during developmental and regenerative osteogenesis at cellular resolution" (p.3, line 61). Should this be the real aim, at the beginning of the results section a rationale why a specific staining for FABP5 was performed, should be included.

5. Does the timing of tamoxifen-induced PDGFR labelling at day 1 to 3 after birth guarantee that only mesenchymal stromal cells are labelled or could other cells (of a different origin) also express these PDGFR at this time point? This should be clarified and discussed.

6. It is not novel that bone regeneration recapitulates features of developmental osteogenesis (p. 11, line 250ff.) Septoclasts could play a role in this process, but the possible role of other cells types needs to be taken into consideration and at least discussed.

7. Co-expression of FABP5 with PDGFR should not be "unexpected" (p.4, line 72), since this co-expression was already demonstrated – together with the pericyte marker NG2 (Bando et al 2018 Histochemistry and Cell Biology)

8. page 4, line 69: the sentence "of sections from in 3-week-old murine femure" should be corrected

9. page 4, line 77: CD68+ osteoblasts "lack expression of FABP5". What is the yellow signal in Fig1c then? Higher magnification would be needed for those areas.

10. page 6, line 127: it should be indicated that PDGFRb+ cells also express GFP, when mentioning the first time, so it is easier to follow when the next sentence states that the target cells were the GFP+ ones.

11. page 7, line 149: the text should refer to Fig 3C (and extended Fig 4C), not to Fig 4C.

12. page 7, line 147: Ctbs is not shown as claimed in Fig "4C" (which should be Fig. 3C).

13. page 7, line 154f: unlike mentioned, no strong expression of MMP13/MMP14 at protein level in FABP5+ cells is shown.

14. page 8, line 172: the text should refer to Extended Figure 4i when mentioning high abundance of FABP5+ and MMP9+ cells.

15. page 35, line 903 Figure legend (g). "According to scRNA-seq data, SCs express MMP9, MMP13, MMP14." This should be also co-localised at protein level by staining.

16. page 9, line 221: the number "14" should be inserted following "post-fracture day".

Figures:

17. Fig. 1C: the figure should indicate the region where the magnifications on the right are from.

18. Fig.1F: What is the yellow labelling in P14? Co-expression of GFP and FABP5? It should be indicated in the Figure legend. Higher magnifications of selected areas would also be required to appreciate the claimed "increase" (in number or in size?) of FABP5+ cells.

19. Fig. 2g: What is the "red" label in the bottom left picture (not mentioned on top of the figure)? Should this picture show only blue and green? The figure should indicate the areas to which the magnifications correspond.

20. Fig. 2H: What is the yellow staining in 2h, if there is no co-staining with hematopoietic cells (=Vav reporter)?

21. Figure 3I: The enlargement (top, middle) does not correspond to the indicated frame in the top left image.

22. Figure 1/4: Was activity of the detected MMPs demonstrated? This could be done by either using antibodies specific for the active MMP or by detecting e.g. DIPEN (aggrecan neoepitope generated following MMP-cleavage).

23. Figure 5f (also Fig 3F): Were hypertrophic chondrocytes only defined by size and morphology? A co-staining with Type X Collagen would verify the hypertrophic character of the chondrocytes.

24. Fig. 6 F/G: Where is the septoclast cluster located? Why is it (in 6G) now defined as the MMP9+ cells, whereas FABP5 expression is not shown? As shown in Fig 3D, also FABP5 negative cells expressed MMP9.

25. Fig. 7H/J: The co-staining of FABP5 and MMPs is rather minimal. What are the other cells, positive for FABP5 but not expressing MMPs and vice versa? The areas corresponding to the magnifications should be indicated.

We would like to thank all reviewers for their time, effort and constructive suggestions, which are greatly appreciated and have enabled us to improve the manuscript further. As you will see, we have included a substantial amount of new data and have also improved the data presentation by showing higher magnification images, single optical planes or isolated channels where appropriate. We trust that the extensive revision has addressed all issues raised by the expert reviewers with the exceptions of a few minor points where a certain antibody staining could not be used and, instead, the conclusion is supported by other data.

REVIEWER COMMENTS

Reviewer #1 (Remarks to the Author):

Sivaraj et al. report that FABP5-expressing septoclasts (SC) at the chondro-osseous border zone are responsible for cartilage matrix degradation and chondrocyte phagocytosis. The authors report that septoclasts have a mesenchymal origin and that interaction with endothelial cells through Notch signaling is important for the identity of septoclasts. The use of several mouse models and numerous scRNA-seq analyses provide novel data, but they are mainly descriptive and sufficient functional prove that septoclasts are the cells that resorb the cartilage matrix is lacking.

Major comments:

1. Typical for osteoclasts is that they secrete not only proteases to cleave proteins, but also produce protons within a sealing zone to dissolve the calcium-phosphate crystals deposited in the matrix. In the present study no information is provided whether septoclasts can also dissolve the calcium-phosphate crystals of the calcified cartilage matrix or that other cell types are still necessary for mineral removal.

We thank the reviewer for these comments. As we mention in the manuscript, expression of vATPase, a critical proton pump for bone resorption, is associated with osteoclasts but not septoclasts. In addition, osteoclasts are associated with the surface of bone structures, whereas septoclasts are concentrated in proximity to chondrocytes of the growth plate or in the fracture callus. However, it has been also reported that osteoclasts are capable of particle/crystal phagocytosis (Wang et al. 1997, PMID: 9227347; Heymann et al. 2001,

PMID: 11193210). In this context it is interesting that we occasionally observe calcium-phosphate crystal-like structures inside SCs by electron microscopy (Supplementary Fig. 2b). In addition, we have added new data showing that culture of SCs in osteo assay surface plates leads to altered cell morphology (Supplementary Fig. 7c) and removal of calcium phosphate (Supplementary Fig. 7d).

Supplementary Fig. 2b (revised version)

Supplementary Fig. 7c, d (revised version)

2. The authors describe that FABP5 is mainly expressed by cells at the chondro-osseous border. However, in Fig. 1a FABP5 staining is also observed in the bone marrow compartment of the metaphysis and diaphysis. Image magnifications of these regions should be provided (separate DAPI-FABP5 channels). Which cell types in the bone marrow are expressing FABP5? These data suggest that FABP5+ cells are a heterogenous population and this heterogeneity will influence the interpretation of the scRNA-seq data. It is therefore crucial to thoroughly describe (and identify) the different FABP5 populations.

As the reviewer points out correctly, FABP5 staining is also observed in sparse cells within the metaphysis and bone marrow. However, the signal is much weaker than in septoclasts (see Fig. 1a-b below and Supplementary Fig. 1a-b). We are now providing better quality tile scan overview images, higher magnifications of critical areas, and show FABP5 expression in a separate channel.

According to our scRNA-seq data, the sparse cells with weak FABP5 immunostaining are mpMSCs in the metaphysis (see Fig. 2c). While anti-FABP5 immunostaining is very useful for visualizing septoclasts in bone sections, the low signals seen in other cells do not at all affect the interpretation of the scRNA-seq data, because cell populations are clustered on the basis of many differentially expressed genes instead of a single marker. In fact, scRNA-seq, unlike bulk sequencing, nicely addresses the issue of cellular heterogeneity and shows that SCs form a distinct population with characteristic properties such as high expression of metalloproteinases.

Figure 1a,b (revised version)

Supplementary Fig. 1a-b

3. The authors describe PDGFR α/β to be expressed by SCs and show co-localization with FABP5. The percentage of FABP5+ cells that express PDGFR α should be quantified in order to show how good the expression-overlap is? This information is important for the interpretation of the scRNA-seq analysis, as part of the FABP5+ SCs population may be not included in the analysis as only PDGFR α -GFP cells were considered for analysis.

As suggested by the reviewer, we have quantified the fraction of *Pdgfra*-GFP+ FABP5+ double positive cells in tissue sections. Of a total of 61.00 ± 3.141 FABP5+ cells analyzed, - 97.38 ± 0.6366 were GFP+ (see Fig. 1c in the revised manuscript) and we are therefore confident that no SC subset has been lost due to our sorting procedure. At the same time, SCs represent a rare subpopulation and are not sufficiently covered by scRNA-seq without prior enrichment.

Figure 1c (revised version)

4. The authors propose that SCs and not osteoclasts (or other cells) are responsible for cartilage degradation during endochondral bone formation. However, the evidence for this remains rather descriptive and not all data are supporting this claim. Fig. 3d shows that MMP9-expressing cells most closely localized to the growth plate are not FABP5+ cells, indicating that another cell type may be responsible for matrix degradation. In addition, the authors claim that the lysosomal protein LAMP1 is detected in septoclasts, but not in osteoclasts. Nevertheless, lysosomes are considered an integral part of osteoclast functioning and Lamp1 has been detected in these cells (Ferron M., *Genes & Development*, 2013), questioning thus the technicality of the staining. Proof that lysosomes are not present in these osteoclasts should be provided. Furthermore, the presence of lysosomes is not sufficient to support the claim that septoclasts are resorbing the cartilage matrix, as lysosomes are present in several cell types simply for proteostasis. In addition, detection of GFP-positivity in FABP5+ cells (Fig. 3g) can be derived from GFP leakage out of the dying chondrocytes and does not prove that septoclasts phagocytose chondrocyte fragments. Therefore, functional proof should be provided to support the claim that septoclasts resorb cartilage matrix. A possible approach to support this claim is to perform in vitro co-culture of chondrocyte cultures/micromasses with sorted SCs or sorted osteoclasts to indeed show that SCs are capable to resorb cartilage.

While we indeed propose that septoclasts mediate matrix degradation and chondrocyte phagocytosis during development and fracture healing, we are not excluding an involvement (and important roles) of other cells such as osteoclasts or endothelial cells. In fact, we are well aware of the existing literature on this subject and discuss relevant publications in our article. Moreover, osteoclasts and SCs can be often seen in proximity (see Supplementary Fig. 1e), raising the possibility of direct or indirect cooperation between these cells.

We agree that MMP9 immunostaining (Fig. 3d) can be seen outside of septoclasts in direct proximity of the growth plate. While this might represent MMP9 protein expression by other cell types, it also needs to be considered that MMP9 is a secreted metalloproteinase so that tissue distribution (immunostaining) is not necessarily identical with expression. In fact, a lot of the MMP9 immunosignal at the chondro-osseous border is not associated with cell bodies (see immunostaining in combination with DAPI in Fig. 3d) and might therefore reflect protein that is either secreted or associated with long cell protrusions.

It is also important to note that our scRNA-seq analysis clearly and consistently shows high expression of *Mmp9* transcripts in SCs relative to other cell types in different data sets (Fig. 2b, c; Fig. 3c; Fig. 4f; Supplementary Fig. 6c). A caveat here is that osteoclasts are not included in these results and, in our experience, it is very difficult (or impossible) to recover these very large and multinucleated cells in single cell approaches using microfluidics (such as 10x Genomics Chromium). In tissue sections, we see MMP9 immunostaining of CD68+ cells (Fig. 3e), which is consistent with the existing literature.

Fig. 3c; Fig. 4f

Fig. 3d (revised version)

Regarding the LAMP1 immunostaining, we state that this is very strong for septoclasts (Fig. 3h) and is thereby useful for the identification of these cells, while vATPase+ multinucleated osteoclasts show comparably weak signal (Fig. 3h). This does, however, not at all imply that LAMP1 is not expressed by osteoclasts and, as the reviewers correctly points out, there is

nice work from the group of Gerard Karsenty showing LAMP1 upregulation by RANKL in cultured osteoclasts (Ferron et al. 2013). In fact, consistent with the important function of the lysosomal system, public scRNA-seq results (<https://tabula-muris.ds.czbiohub.org>) show very widespread *Lamp1* expression in many different organs and a wide range of cell populations (see below). In conclusion, we use the strong anti-LAMP1 immunostaining seen for SCs mainly as another marker for the identification of these cells, but it is also consistent with a role of septoclasts as resorptive cells.

Regarding the issue of potential GFP “leakage”, we see large *Acan-CreERT2*-labeled particles/structures accumulated in SCs and not in adjacent CD68+ osteoclasts (see Fig. 3g and Supplementary Fig. 6h), arguing against an unspecific process such as diffusion of leaked protein.

Fig. 3g

Supplementary Fig. 6h

As suggested by the reviewer, we performed co-culture experiments with chondrocytes and SCs (Supplementary Fig. 7f). In these conditions, however, chondrocytes were not hypertrophic so that we could not address the endocytosis of cell fragments by SCs. Interestingly, we saw profound induction of MMP9 expression and increased actin stress fibers at the interface between co-cultured SCs and CHOs (Supplementary Fig. 7a, b). This result suggests that some features of SCs are induced by the proximity of chondrocytes.

Supplementary Fig. 7a, b (revised version)

5. The authors inactivate Dll4 specifically in ECs to study the importance of Notch signaling for SC control via Hey1-GFP reporter mice. However, scRNA-seq data in Fig. 4h show that Hey1 is also expressed in ECs and mpMSCs. Therefore, Dll4 deletion can also affect the other Hey1-expressing cell types. The authors should provide proof that the observed phenotype of increase in hypertrophic chondrocytes is solely due to impaired SCs function? In addition, it is known that MMP9 is also produced by ECs: is the observed decrease in MMP9 due to decreased SC or EC function?

In our scRNA-seq data, *Hey1* transcripts are highly expressed in septoclasts, metaphyseal MSCs (mpMSCs) and arterial ECs (Fig. 4g). Arteries are mainly located in the marrow and are therefore unlikely to have direct interaction with chondrocytes. mpMSCs are also positioned in considerable distance from the growth plate and, given that there is presumably no direct link between bone mesenchymal stromal cells and chondrocyte resorption, it is highly unlikely that altered Hey1 expression in these cells is relevant for the observed phenotype. We therefore attribute the increase in hypertrophic chondrocytes to the loss of FABP5+ SCs and substantial reduction of MMP9 expression (Fig. 5e). This is consistent with our scRNA-seq data and immunostaining data indicating high expression of MMP9 and other metalloproteinases in SCs.

Fig. 4f, g

Furthermore, we provide additional *in vitro* results linking Notch signaling to SC function in the revised manuscript. Stimulation of cultured SCs with immobilized recombinant Dll4 led

to increased expression of the Notch target genes *Hey1*, *Heyl* and *Hes1* but we also significant increases in *Fabp5* and *Mmp9* transcripts (Fig. 4k).

Fig. 4k (revised version)

6. In Fig. 5d, the authors quantify the number of SCs. In Fig. 1c, SCs were counted based on FABP5 expression, but in Fig. 5d a difference is made between SC and FABP5 in SC. How are the SC identified if not on FABP5 expression and how are these quantifications performed?

As we show above, *Fabp5* expression in cultured SCs is increased by stimulation with immobilized recombinant Dll4. This is consistent with our *in vivo* observations in EC-specific Dll4 mutants, where *Hey-GFP* expression, the number of FABP5+ SCs and the levels of FABP5 expression are reduced (red signal, Fig. 5c). The lower FABP5 expression was still sufficient to determine the number of positive cells (Fig. 5d).

Fig. 5c-d

7. The authors report that SC number and MMP9 expression are decreased upon *Dll4* deletion. To prove that SCs are responsible for cartilage resorption during bone development and fracture healing, the authors should provide Safranin O or Collagen2 staining to investigate whether cartilage remnants are observed, which would indicate defective cartilage resorption in *Dll4* mutants? This experiment would provide additional evidence that SCs are responsible for cartilage degradation.

As suggested by the reviewer, we performed Safranin-O staining to show the striking expansion of the growth plate in EC-specific *Dll4* mutant bone. This shows that the expansion can be attributed to the increase of hypertrophic chondrocytes. Furthermore, one can see cartilage remnants extending from the growth plate into the adjacent metaphysis, which is strongly increased in EC-specific *Dll4* mutants (Supplementary Fig. 9d).

We were having some issues with the specificity of the Collagen type II and X immunostaining therefore not including them in the revised manuscript. But even without these results it is clear that defective cartilage resorption is a major issue in EC-specific *Dll4* mutants.

Supplementary Fig. 9d (revised version)

Minor comments:

1. Fig. 1d. The identification of the different cell types by electron microscopy looks rather arbitrary as no information is provided what the basis was for identifying some cells as being septoclasts. A similar comment applies to fig 3i where it is not clear on which basis cells are considered to be SC or OC. In addition, the LAMP1 labeling of Lamp1 vs vATPase is not clear.

Based on finding published by Lee et al. 1995 (PMID: 7730591) and our own observations, we define criteria for the identification of SCs in electron micrographs in the manuscript. These include location near growth plate chondrocytes, distinct elongated morphology, strong actin filaments, abundance of vesicular structures, and a highly characteristic enlargement of the rough endoplasmic reticulum (similar to plasma cells). These features distinguish SCs from other cell types in the same region, namely chondrocytes, osteoclast, osteoblast, and endothelial cells. Regarding the LAMP1 staining, this was chosen based on our immunofluorescence staining (Fig. 3h, left column). We appreciate that the small immunogold labels are not easy to see without substantial magnification, but this is also the reason for placing differently colored arrowheads.

2. Ext. Fig. 2a. FABP5+ cells are described as being highly proliferative, but no quantification is performed and the images show only a few EdU -positive FABP5+ cells.

We have added new confocal images and quantification, as requested by the reviewer. This shows that SCs proliferate more than other *Pdgfra-GFP*+ cells.

3. Fig. 2b-c: dpMSCs and mpMSCs are described as different cell populations based on respectively *Esm1* and *Postn* expression. The authors should provide evidence that these markers are specific for stromal cells in either the diaphyseal or metaphyseal region respectively.

First of all, it is important to emphasize that the clustering of subpopulations in the scRNA-seq analysis is done on the basis of many genes, a selection of which is shown in Fig. 4c and Supplementary Fig. 4d. The detailed characterization of bone stromal mesenchymal cell subpopulations, including dpMSCs and mpMSCs, is the main subject of a separate study, which has just been published (PMID:34260921). In this study, which does not mention the topic of septoclasts, we show that *Esm1* staining labels dpMSCs, whereas *Hey1-GFP* expression marks mpMSCs (see below). We did not pursue periostin immunostaining because this is a secreted protein, which is strongly expressed in a regional pattern such as in the activated periosteum (PMID: 29472541) and is therefore of limited use for the visualization of individual cells.

4. Fig. 2a and Ext Fig. 3h: since it is claimed that septoclasts are not derived from the hematopoietic lineage it is important to show images with localization of GFP and FABP5+ cells (red) without any other staining. Now the 3 stainings are showed together (GFP, FABP5 and CD68). Especially, in the bottom panels, large green cells (clusters) are observed almost at the same location as FABP5+ cells.

We agree with the reviewer and have updated the images and show *Vav-Cre*-mediated GFP expression and FABP5 immunostaining cells at high magnification and with single channel images (Fig. 2h, Supplementary Fig. 5c-f). Furthermore, we have extended the analysis of the BMSC marker PDGFR β and the osteoclast markers CD68 and vATPase in *Vav-Cre-mTG* bone.

Fig. 2h (revised version)

Supplementary Fig. 5c-f (revised version)

5. Fig. 3b and Extended Fig. 4a-b: information should be provided whether GSEA data show the top-ranking upregulated pathways or whether a selection of pathways of interest was made for the visualization? Please include this information in the figure and/or text.

Thank you for pointing this out. GSEA results show genes with 2-fold higher expression in SCs relative to BMSCs. Top-ranking upregulated pathways are displayed in Fig. 3b and Supplementary Figure 6a, b.

6. Figure 6: scRNA-seq is performed on control and fractured bone, but no information is provided which region of the fractured and control bone was used. This information is important as chondrocytes were also detected in the control condition.

We used whole femurs (including epiphysis, growth plate, metaphysis and marrow) for control and fracture samples in the scRNA-seq analysis.

Reviewer #2 (Remarks to the Author):

This paper addresses with the most modern techniques the cellular origin of the septoclasts, a cell type popularized mainly by the Amizuka group and that is supposed to resorb cartilage. Resorption of cartilage is critical during embryonic development to initiate ossification in bone forming through endochondral ossification. The main contribution of this paper which is quite important for the field is to show that septoclasts are of mesenchymal origin. The demonstration here is very convincing. There are however several concerns that need to be addressed. Some of them are experimental others in the interpretation of the results.

Most experiments presented are done after birth and often in 3 week-old mice. Yet, as the authors present in the introduction, resorption of cartilage during development is absolutely necessary for endochondral ossification to proceed. Therefore what is missing from the paper is the demonstration that FABP5⁺ septoclasts can be seen later during embryonic development when cartilage is resorbed and ossification actually starts. Therefore, data that are presented in Figure 1a needs also to be shown in E17.5 or E18.5 embryos.

We thank the reviewer for the kind assessment and the constructive feedback. We show that FABP5⁺ SCs can be observed in chondro-osseous border zone during embryonic development. As in postnatal stages, SCs are more abundant than osteoclasts in the proximity of cartilage, consistent with a role in chondrocyte resorption, whereas osteoclasts are found throughout the primary ossification center (Fig. 1f and Supplementary Figure 2g, h).

Fig. 1f

Supplementary Fig. 2g, h

- FAB5+ septoclasts are present in the cortical bone (Figure 1a), a fact that is not mentioned by the authors. What may be the function of these cells there? do they express the same genes that close to the growth plate?

As mentioned in one of our replies to reviewer #1, FABP5 staining is also observed in sparse cells within the metaphysis and bone marrow. However, the signal is much weaker than in septoclasts (see Fig. 1a-b below and Supplementary Fig. 1a-b). We are now providing better quality tile scan overview images, higher magnifications of critical areas, and show FABP5 expression in a separate channel.

There is, however, no signal in cortical bone at least in our staining's (see bottom panels in Supplementary Fig. 1b).

Supplementary Fig. 1a-b

- Another question raised by the data presented in Figure 1b is why do the number of septoclasts is lower at P1 than at p14? Also what happens between P14 and P28 Figure3j when their number begins to decline?

This is a good question. The number of SCs increases during postnatal development and declines as soon as bone grows slows down. In the literature, it has been shown that growth plate size gradually declines after P6 but the size (height) of the hypertrophic zone peaks at P13 before it declines (PMID: 31997656). While the stages investigated in our study diverge by 1 day, there is a nice correlation between the abundance of SCs and hypertrophic chondrocytes. Moreover, the growth rate of femur and other long bones strongly declines after P16 (PMID: 7649813), which is consistent with the reductions in hypertrophic

chondrocytes and SCs mentioned above.

Fig. 1g (revised version)

- An important question that needs to be addressed is to know if septoclasts are present or absent in bones that form through intramembranous ossification. This should not be difficult to achieve.

We agree that this is a good question. Consistent with the absence of chondrocytes, FABP5+ cells are rare in 3-week-old calvaria and are associated with vessels penetrating the cranial bone (Supplementary Fig. 3d, e). These data indicate that FABP5+ cells in bone are predominantly associated with endochondrous osteogenesis. We also mention this in the revised text.

Supplementary Fig. 3d-e (revised version)

- The authors show clearly that septoclasts express at high levels MMPs (Figure3) and this is unquestionable, however there is only a correlation between the expression of the MMPs and Lamp1 in FAB5+septoclasts and the fact they may be resorbing cartilage. The demonstration has not been made in this paper. This is fine but the title of the paper, and throughout the results section should be rewritten in a more cautious manner.

We show that SCs express high levels of various metalloproteinases and also internalize fragments of genetically labelled chondrocytes *in vivo*. While we indeed propose that SCs resorb cartilage, we are not claiming that these are the only cells endowed with this function. SCs are located in close proximity of endothelial cells and osteoclasts and might cooperate with these cell populations. We also mention the existing literature on endothelial cells and osteoclasts in cartilage resorption in the manuscript and have carefully avoided any unsubstantiated claims.

- There is a quantification of the number of septoclasts in Figure 2 but not in Figure 4, could the authors provide this quantification?

We are not sure whether we understand this question. If it was about Fig. 1 (not 2) and Fig. 4, we have included additional stage in Fig. 1g, which now also covers later postnatal stages (shown in Fig. 4).

- In the last 2 figures the authors address whether septoclasts are present during fracture healing. They show clearly that it is the case in a bone that develop through endochondral ossification. A control of specificity is to show if it is the case or not in fracture healing in bones developing through intramembranous ossification.

As mentioned in our reply to your previous question, we now show that FABP5+ SCs are rare in the calvarium and are therefore unlikely to play a role in intramembranous ossification. Moreover, we show that MMP9 expression by SCs *ex vivo* is induced by contact with chondrocytes (Supplementary Fig. 7a, b), which suggests that important features of septoclasts might get induced by the proximity of chondrocytes.

- What the authors show that is new in Figure 7 is that septoclasts re-emerge during fracture repair. They don't show "that regenerating bone recapitulates major features of developmental osteogenesis" because that is already well established. The authors need to also qualify this statement and stay closer to the data.

We do not question that other studies have identified similarities between developmental and regenerative osteogenesis, but two important aspects, namely the mode of (budding) angiogenesis – as opposed to sprouting angiogenesis - and the reemergence of septoclasts have not been addressed previously. We have checked the wording of our text to ensure that all our conclusions are covered by our results.

--

Reviewer #3 (Remarks to the Author):

General comments:

In this study, the authors identify septoclasts, defined by their expression of FABP5, to be responsible for chondrocyte phagocytosis and matrix degradation in the chondro-osseous border of the growth plate. They further demonstrate that these septoclasts are of mesenchymal origin – unlike osteoclasts – by lineage tracing. Septoclast specification is claimed to be mediated by endothelial cell via notch signalling. While septoclasts disappear with time after development, they re-emerge for bone repair.

The main novelty of the presented work is the identification that such septoclast population derive from stromal/mesenchymal progenitors (in contrast to hematopoietic-derived osteoclasts). This is investigated by extensive immunofluorescent stainings, lineage tracing and combined scRNA sequencing. However, the authors do not convincingly distinguish the identified cell population from perivascular cells and, while the mesenchymal origin is demonstrated for some cells, hematopoietic origin of at least some septoclasts cannot be entirely excluded.

We thank the reviewer for this assessment and trust that the extensive revisions will address all questions and concerns.

Major issues:

1. The here shown septoclasts display several similarities with perivascular cells, such as the location, the mesenchymal origin, PDGFRB expression and their re-activation upon injury. Indeed, former studies suggest septoclasts to originate from pericytes, due to such co-expression (Bando 2018 Histochemistry and Cell Biology). Here the authors, however, conclude, that septoclasts are not pericytes because "...unlike pericytes, they are not embedded in the subendothelial basement membrane" p.4. line 86f. In order to distinguish clearly septoclasts from pericytes, better characterisation is required, such as staining for the expression of pericyte markers (e.g. NG2, CD146) and for nestin.

We thank the reviewer for this comment. After having worked on pericytes and blood vessels in many different organs for nearly 25 years, I am still finding the identification of pericytes in bone very challenging for numerous reasons. As is nicely illustrated in a seminal review by my colleague and leader in the pericyte field Christer Betsholtz (PMID: 21839917), there are no pericyte-specific markers and all known markers that are frequently used in the literature are shared by other cell types including smooth muscle cells, certain fibroblasts and many mesenchymal stromal cells. This notion has been recently supported further by single cell RNA sequencing analysis of mural cells from different organs. Embedding in the subendothelial basement membrane is a morphological hallmark of pericytes that is also described in the review by Betsholtz but also in many other publications. In bone, many mesenchymal cell-derived cell populations express some markers that are characteristic for mural cells and are tightly associated with blood vessels, as is the case for reticular cells in marrow or mesenchymal stromal cells in the metaphysis. Arguing that blood vessels bone might even lack "classical" pericytes, a recent scRNA-seq study from my group (PMID:34260921) shows a distinct vascular smooth muscle cell population in bone, whereas the same approach did not recover pericytes as we readily find them in other organs. Other scRNA-seq studies analyzing bone stromal cells in the adult have also not reported a distinct pericyte population (PMID: 30971824; PMID: 31871321). In contrast, the group of David Scadden describes a pericyte population in their scRNA-seq data, but it should be noted that these cells express alpha-smooth muscle actin (*Acta2*), which is a marker of vascular smooth muscle cells but not pericytes in adult brain and lung

(<http://betsholtzlab.org/VascularSingleCells/database.html>). It might be also possible that the bone vasculature does not require pericytes because of the modest blood flow velocity (PMID: 27922003, PMID: 28199850) or because other mesenchymal cell-derived stromal cell populations fulfill this function. As there is no conclusive data in one or the other direction and because this question is also not the subject of the current manuscript, we are not discussing the somewhat elusive nature of bone pericytes or their relationship with septoclasts. We merely make the statement about septoclasts that “unlike pericytes, they are not embedded in the subendothelial basement membrane”, which is supported by our data. As requested by the reviewer, we analyzed the expression of additional markers, namely NG2 and CD146, which are frequently expressed by pericytes in other organs (but also other cell populations). Broad expression is certainly a feature of NG2, which labels many different cell populations in bone (Supplementary Fig. 2d). The same pattern can be seen with a genetic reporter (Tg(Cspg4-DsRed.T1)1Akik/J; PMID: 18045844). Consistent with previous reports about CD146 expression in endothelial cells (PMID: 11739172), immunostaining prominently labels EMcn+ endothelial cells and arteries but not vessel-associated cells in proximity of the growth plate (Supplementary Fig. 2e). As we do not think that the inclusion of additional markers would conclusively address the issue of bone pericytes nor their relationship with septoclasts, we have not performed further marker analysis. In our view, this question is better left to future studies using clear-cut genetic lineage tracing or other approaches leading to conclusive results.

Supplementary Fig. 2d-e (revised version)

2. While the mesenchymal origin of FABP5+ cells is demonstrated by deriving from

PDGFRB labelled cells, it looks like some FABP5+ cells also express CD68/Vav. This could indicate that at least a fraction of these cells may be of hematopoietic origin (“yellow” cells in Fig. 1C and Fig 2h). This should be clarified and discussed.

We think that this impression might be caused by the use of thick cryosections for our immunostainings. These thicker sections facilitate the imaging of 3D structures by confocal microscopy, such as blood vessels, but they may also falsely suggest that stained cells in different optical planes are co-localized. This issue is aggravated by the fact that different cell types, including osteoclasts, ECs and SCs, are often located in direct proximity. We now provide additional images showing higher magnifications and single optical planes, which clearly show that osteoclasts but not SCs are derived from *Vav-Cre*-labelled hematopoietic cells.

Fig. 2d (revised version)

Separated channels

Fig. 2h

3. The here presented data related to notch signalling mediated septoclast activation are rather descriptive and lack the functional prove, that septoclast activation is regulated by endothelial cells via *dll4* activating the notch pathway. Could their “disappearance” in *Dll4* mutants also result from physical removal of endothelial cells and/or as a consequence of generally reduced vascularisation? To demonstrate direct involvement of notch signalling on septoclast activation, *in vitro* studies with isolated septoclasts should be performed, in which activation and inhibition of the notch signalling pathway can be demonstrated in a cleaner experimental setting. Additional co-culture experiments with endothelial cells expressing or not *Dll* could further imply a possible direct interaction between these cell types.

The here provided data so far do not support the statement in the discussion (p11, line 268f) that “Septoclasts are also controlled by Notch signalling through interactions with *Dll4*+ vessel buds at the chondro-osseous interface in the fracture callus”

We agree with the reviewer and have modified the sentence in the discussion accordingly. To provide better insight into the regulation of SCs by Notch signaling, we have performed *ex vivo* stimulation experiments of SCs with immobilized recombinant *Dll4*. This shows that Notch activation induces the expected expression of Notch target genes but also of *Fabp5* and *Mmp9* transcripts (Fig. 4k). These findings are consistent with our *in vivo* data showing significant reductions in SC number but also in FABP5 and MMP9 expression in EC-specific

Dll4 mutants (Fig. 5d, e). It is also clear that blood vessels are still present at the chondro-osseous interface in *Dll4* mutants, arguing that the SC defects are indeed caused by the loss of *Dll4* and not by the absence of ECs (Fig. 5e).

Additional comments:

4. The aim of the study is rather generic: “..systematically investigate the processes during developmental and regenerative osteogenesis at cellular resolution” (p.3, line 61). Should this be the real aim, at the beginning of the results section a rationale why a specific staining for FABP5 was performed, should be included.

We have performed a series of different immunostainings early in the project and have modified the first sentence of the results accordingly.

5. Does the timing of tamoxifen-induced PDGFR labelling at day 1 to 3 after birth guarantee that only mesenchymal stromal cells are labelled or could other cells (of a different origin) also express these PDGFR at this time point? This should be clarified and discussed.

We have just published a detailed characterization of *Pdgfrb-CreERT2*-labelled cells involving scRNA-seq and genetic fate tracking, which shows that this line initially labels mesenchymal stromal cells in the metaphysis, which later give rise to other mesenchymal cell populations including reticular cells in marrow and adipocytes (PMID:34260921). In the current manuscript, we show that septoclasts but not osteoclasts are labelled by *Pdgfrb-CreERT2*, whereas the opposite is true for *Vav-Cre*, which labels hematopoietic cells and their progeny, including osteoclasts.

6. It is not novel that bone regeneration recapitulates features of developmental osteogenesis (p. 11, line 250ff.) Septoclasts could play a role in this process, but the possible role of other cells types needs to be taken into consideration and at least discussed.

Agree. This issue has been also mentioned by reviewer #2. We do not question that other studies have identified similarities between developmental and regenerative osteogenesis, but two important aspects, namely the mode of (budding) angiogenesis – as opposed to sprouting

angiogenesis - and the reemergence of septoclasts have not been addressed previously. We have checked the wording of our text to ensure all our conclusions are covered by data. While we indeed propose that septoclasts mediate matrix degradation and chondrocyte phagocytosis during development and fracture healing, we are not excluding an involvement (and important roles) of other cells such as osteoclasts or endothelial cells. In fact, we are well aware of the existing literature on this subject and discuss relevant publications in our article.

7. Co-expression of FABP5 with PDGFR should not be “unexpected” (p.4, line 72), since this co-expression was already demonstrated – together with the pericyte marker NG2 (Bando et al 2018 Histochemistry and Cell Biology)

We have modified this sentence, as requested by the reviewer.

8. page 4, line 69: the sentence “of sections from in 3-week-old murine femure” should be corrected

Thank you for alerting us to this mistake, which we have corrected.

9. page 4, line 77: CD68+ osteoblasts “lack expression of FABP5”. What is the yellow signal in Fig1c then? Higher magnification would be needed for those areas.

As mentioned above, we use thick cryosections for our immunostainings, which facilitates the imaging of 3D structures by confocal microscopy, but may also falsely suggest that stained cells in different optical planes are co-localized. We have addressed this issue by providing more high magnification images and separated channels.

Fig. 2d (revised version)

10. page 6, line 127: it should be indicated that PDGFRb⁺ cells also express GFP, when mentioning the first time, so it is easier to follow when the next sentence states that the target cells were the GFP⁺ ones.

We have included a short statement, as requested by the reviewer.

11. page 7, line 149: the text should refer to Fig 3C (and extended Fig 4C), not to Fig 4C.

Thank you for alerting us to this mistake, which we have corrected.

12. page 7, line 147: Ctbs is not shown as claimed in Fig “4C” (which should be Fig. 3C).

Thank you for alerting us to this mistake, which we have corrected.

13. page 7, line 154f: unlike mentioned, no strong expression of MMP13/MMP14 at protein level in FABP5⁺ cells is shown.

Thank you for alerting us to this point. Unfortunately, the antibodies detecting FABP5 and MMP13 or MMP14 cannot be combined. However, like MMP9, MMP13 immunosignal is also concentrated at the chondro-osseous border zone where SCs are abundant, which is consistent with our scRNA-seq showing that transcripts for multiple metalloproteinases are highly expressed in SCs. Moreover, FABP5⁺ SCs show very high levels of anti-MMP9

immunostaining (Fig. 3d) and there is substantial co-localization of MMP9 and MMP13 (Fig. 3e). We have modified the wording of the text and figure legends to reflect this issue.

14. page 8, line 172: the text should refer to Extended Figure 4i when mentioning high abundance of FABP5+ and MMP9+ cells.

Thank you for alerting us to this point, which has been addressed as part of this revision.

15. page 35, line 903 Figure legend (g). “According to scRNA-seq data, SCs express MMP9, MMP13, MMP14.” This should be also co-localised at protein level by staining.

As mentioned above, the antibodies detecting FABP5 and MMP13 cannot be combined. However, like MMP9, MMP13 immunosignal is also concentrated at the chondro-osseous border zone where SCs are abundant, which is consistent with our scRNA-seq showing that transcripts for multiple metalloproteinases are highly expressed in SCs.

16. page 9, line 221: the number “14” should be inserted following “post-fracture day”.

Agree. This has been corrected.

Figures:

17. Fig. 1C: the figure should indicate the region where the magnifications on the right are from.

Agree. The region has been indicated (now Fig. 1d).

18. Fig. 1F: What is the yellow labelling in P14? Co-expression of GFP and FABP5? It should be indicated in the Figure legend. Higher magnifications of selected areas would also be required to appreciate the claimed “increase” (in number or in size?) of FABP5+ cells.

Given that FABP5+ SCs and EMCN+ ECs are located in close proximity, the immunostaining of thick cryosections might give the impression of co-staining in maximum intensity projections (merging of all optical planes). However, the analysis of single optical planes shows that there is no co-expression of the two markers.

We see an increase in the number of SCs during early postnatal development and have clarified this point in the revised manuscript.

19. Fig. 2g: What is the “red” label in the bottom left picture (not mentioned on top of the figure)? Should this picture show only blue and green? The figure should indicate the areas to which the magnifications correspond.

We have updated the labels to make sure that all signals/markers are shown in the corresponding panels. For the higher magnification images in the bottom row, we now indicate the corresponding area in the overview image.

20. Fig. 2H: What is the yellow staining in 2h, if there is no co-staining with hematopoietic cells (=Vav reporter)?

As mentioned above, the close proximity of different cell types and cellular processes might generate the impression of co-staining in thick cryosections, which we use to capture 3D structure. We now provide more higher magnification images, single optical planes and isolated channels to address this issue.

As you can see below (Fig. 2e and Supplementary Fig. 5e, f), *Vav-Cre*-labelled (GFP+) osteoclasts can be found in direct proximity of FABP5+ septoclasts, generating some yellow overlap at the contact area between the cells.

21. Figure 3I: The enlargement (top, middle) does not correspond to the indicated frame in the top left image.

Agree. We have corrected the position of the boxed areas.

22. Figure 1/4: Was activity of the detected MMPs demonstrated? This could be done by either using antibodies specific for the active MMP or by detecting e.g. DIPEN (aggrecan neoepitope generated following MMP-cleavage).

Unfortunately, we were unable to get hold of these antibodies. However, it has been shown that anti-DIPEN immunostaining decorates the chondro-osseous border in the tibiofemoral

joint of 3-week-old wild-type mice (PMID: 15798221). The same study, however, also proposes that matrix metalloproteinases are not essential for aggrecan turnover.

23. Figure 5f (also Fig 3F): Were hypertrophic chondrocytes only defined by size and morphology? A co-staining with Type X Collagen would verify the hypertrophic character of the chondrocytes.

We have added a Safranin-O staining (Supplementary Fig. 9d), which shows the expected organization of hypertrophic chondrocytes in control animals and their accumulation in EC-specific *Dll4* mutants.

24. Fig. 6 F/G: Where is the septoclast cluster located? Why is it (in 6G) now defined as the MMP9+ cells, whereas FABP5 expression is not shown? As shown in Fig 3D, also FABP5 negative cells expressed MMP9.

We are now providing improved trajectory analysis and other bioinformatics data to address these questions. It has to be noted *Fabp5*+ cells represent a small population that clusters with mpMSCs (Figure 7a, b), which is different from experiments using enrichment approaches and genetic markers (see Fig. 2a-c). We now indicate the position of the *Fabp5*+ in the trajectory plots (see red arrow in Fig. 7b). In the absence of the trajectory line (see further below), the overlap between *Fabp5* and *Mmp9* in the indicated position (green cells) is more easily appreciated.

Figure 7a, b (revised version)

Without trajectory line

Furthermore, we have now extended the analysis of BMSCs and, in particular the mpMSC subcluster, which enables the separation of SCs from these cells (Fig. 7f-h). These results also nicely show the increased frequency of *Fabp5*+ SCs in fracture conditions relative to control samples and increased in *Mmp9* and *Fabp5* expression in these cells (Fig. 7f-h and Suppl. Fig. 10e, f).

Fig. 7 f-i (revised version)

Supplementary Fig. 10e, f

25. Fig. 7H/J: The co-staining of FABP5 and MMPs is rather minimal. What are the other cells, positive for FABP5 but not expressing MMPs and vice versa? The areas corresponding to the magnifications should be indicated.

FABP5 immunostaining labels the cell body and its processes because the protein is located in the cytosol and plasma membrane. In contrast, MMP9 and other metalloproteinases are secreted proteins and therefore not all the immunosignal is associated with cell bodies, which can be best seen in Fig. 3d (for postnatal development), where DAPI-stained nuclei are included, but also in Fig. 7g-j (fracture conditions) in the revised manuscript (see below). Areas corresponding to the magnifications are now indicated in Fig. 7g and 7h.

Fig. 7g-j

Reviewers' Comments:

Reviewer #1:

Remarks to the Author:

The authors responded to many comments and questions by providing additional data and adapting the text where necessary.

Some (small) issues still remain.

Fig. 1f: the quantification of SC and OC is performed on E17.5 sections, but no osteoclasts are shown at this age, as Suppl Fig 2g shows CD68 pos cells at E16.5.

Fig. 1g: for the statistical analysis, one-way anova should be considered instead of multiple t-tests.

Fig. 1h: the scheme is somewhat misleading, as SC are the only cells shown in contact with the growth plate and osteoclasts are only present at the bone surface, while the authors show that osteoclasts are also present at the growth plate boundary, although less abundant (around 30% of SC at 3 wks and 25% of SC at E17.5)

Fig 3e: it is not clear how the authors can state that osteoclasts express lower levels of MMP9 compared to SC (line 158), when identification of SC and OC by staining is performed on different sections.

Supplementary Fig 7a-d: several in vitro experiments were performed to confirm the resorptive capacity of the SC, but several questions remain. In the chondrocyte-septoclast coculture, it is rather surprising that phalloidin staining is very weak in the septoclasts, as gene expression data show enrichment for cytoskeleton organization in these cells. In addition, it is not clear what the localization is of the SCs after 5 days related to the chondrocytes and it seems that MMP9 is expressed by the chondrocytes and not by the SCs, although the authors state that MMP9 is expressed in SCs (line 178). Proper validation of the type of cell that produce MMP9, and other MMPs (MMP13, MMP14), is necessary, as chondrocytes can also increase MMP9 production in stressed conditions. The septoclasts seem to release some coated Ca-P, although this level seems very limited when you compare it to osteoclasts cultures in other studies. Further proof of the resorptive capacity of the SC should therefore be provided.

Reviewer #2:

Remarks to the Author:

The authors should be commended for having done a large amount of work to address the concerns of the reviewers. My own concerns are been addressed satisfactorily.

Reviewer #3:

Remarks to the Author:

General comments:

The authors have performed extensive additional analyses and included new images of improved quality to underline their statements. However, they replied to the comments and issues raised by the reviewers in an only partially satisfactory way.

As a general comment, the reviewer would appreciate if the changes performed in the revised version could be highlighted for easier navigation through the manuscript and better appreciation of the performed additional analyses and their discussion.

Specific comments to authors responses to reviewer 3:

#1. As requested, additional staining for pericyte markers was performed and implemented as Supplemental Figure 2 d,e.

#2. Although claimed to be "additional", the revised Fig 2d seems identical to the previous Fig 2c. In addition, it remains unclear where the newly added single optical plane figure (from the rebuttal) can be found in the manuscript.

#3. As requested, the authors have performed additional ex vivo experiments with Dll4 now clearly showing activation of notch signalling in SCs (Fig 4k). These results now support the authors' statement of the importance of notch signalling in septoclasts. Unlike claimed in the rebuttal, however, the content of sentence of the first submission (p11, 268: "Septoclasts are also controlled by Notch signalling through interactions with Dll4+ vessel buds at the chondro-osseous interface in the fracture callus") was not changed in the revised manuscript (p.12, 291 "..septoclasts are controlled by Notch signalling through interactions with Dll4+ vessel buds at the chondro-osseous interface").

#4. The authors did not consider the comment of the reviewer that the aim of the study was rather generic: "..systematically investigate the processes during developmental and regenerative osteogenesis at cellular resolution" (p.3, line 61 in first submission). The stated aim is still identical in the revised version: "..systematically investigate the processes during developmental and regenerative osteogenesis at cellular resolution (p. 3, line 61).

The reviewer asked to at least provide a rationale in the first sentence of the results, why instead of the proposed systematic investigation only a specific staining for FABP5 was performed. This was however not included. The authors only newly mention that they performed immunostaining "with a range of markers". However, the indicated figures (Fig 1a,b; Suppl Fig 1 a-c) still only show FABP5 staining.

#5. The authors convincingly replied to the question on the timing of tamoxifen, however this should also be mentioned in the manuscript (as previously requested by the reviewer).

#6. Whether the authors have "...checked the wording of our text to ensure all our conclusions are covered by data" is difficult to judge, due to the lack of track changes/highlighted modification.

#7. The indicated grammatical mistake "from in" was not corrected, unlike claimed. (p. 4, line 68).

#9. "We have addressed this issue by providing more high magnification images and separated channels." However, the original Fig. 2c seems identical to the newly claimed Fig. 2d (except the newly added square). See comment #2

#19. The newly included square in Fig. 2g should be mentioned in the legend. It is still rather misleading to see the overlay of the different stainings on the bottom left with the "heading": GFP EMNC. This should be clarified (what is the bottom row?) in the figure and/or legend.

#23. The newly added SaFO definitely shows chondrocytes of a hypertrophic morphology. However, the COL X assessment that was asked for by the reviewer was not performed.

Additional comment(s):

Supplemental Figure: Please, indicate the mentioned area with "electron dense particles" with an arrow. What are the 3 different pictures? What is the difference between the left and the middle one? The right picture seems to be a higher magnification from the middle one. If so, please indicate the area (in the middle one). Also, the scale bar in the right figure seems to indicate a wrong scale, since it is identical to the one in the other figures, although showing higher magnification.

We thank the reviewers for their helpful comments and suggestions. While a detailed point-by-point response to each question is provided below, we would like to clarify one important point. While our work shows that septoclasts contribute to the resorption of chondrocytes via the uptake of chondrocyte debris and the expression of secreted proteases, it is unlikely that the same applies to the resorption of calcified bone. In fact, SCs are concentrated near the growth plate and, unlike OCs, not on the surface of calcified bone. SCs also do not express vATPase, which is critical for pH changes during bone resorption by osteoclasts.

Accordingly, the following statement has been added to the Discussion: “*Unlike osteoclasts, SCs are not associated with calcified bone and show no expression of vATPase, which is critical for local pH changes allowing the dissolution of calcium phosphate. This argues that septoclasts are unlikely to mediate the resorption of calcified bone.*”

REVIEWER COMMENTS

Reviewer #1 (Remarks to the Author):

The authors responded to many comments and questions by providing additional data and adapting the text where necessary.

Some (small) issues still remain.

Fig. 1f: the quantification of SC and OC is performed on E17.5 sections, but no osteoclasts are shown at this age, as Suppl Fig 2g shows CD68 pos cells at E16.5.

Thank you for pointing this out. This quantification is for E16.5 (Suppl. Fig 2g). This is now indicated in the figure.

Fig. 1g: for the statistical analysis, one-way anova should be considered instead of multiple t-tests.

We agree and have updated the statistical analysis to one-way Anova.

Fig. 1h: the scheme is somewhat misleading, as SC are the only cells shown in contact with the growth plate and osteoclasts are only present at the bone surface, while the authors show that osteoclasts are also present at the growth plate boundary, although less abundant (around 30% of SC at 3 wks and 25% of SC at E17.5)

It is obviously not our intention to mislead and we repeatedly state in the manuscript that OCs are found near SCs in proximity of the growth plate. As suggested by the reviewer, we have added osteoclasts (OC) in the graphical summary (Fig 1h).

Fig 3e: it is not clear how the authors can state that osteoclasts express lower levels of MMP9 compared to SC (line 158), when identification of SC and OC by staining is performed on different sections.

It is correct that we do not show SCs and OCs in the same section/image because the relevant antibodies cannot be combined in a single staining. While we have noted that FABP5+ cells show very strong anti-MMP9 signal, it is fair to say that a direct side-by-side comparison with OCs is lacking. Moreover, given that MMP9 is secreted, it is unclear whether cell-associated staining is a valid representation of the production and release of the metalloproteinase. Due to these issues, we have amended the description of this result on page 7 of the manuscript: “*Consistent with the findings above, MMP9 immunosignal decorates FABP5+ cells in the chondro-osseous border zone as well as OCs in the same region (Fig. 3d, e and Supplementary Fig. 6d, e).*”

Supplementary Fig 7a-d: several in vitro experiments were performed to confirm the resorptive capacity of the SC, but several questions remain. In the chondrocyte-septoclast coculture, it is rather surprising that phalloidin staining is very weak in the septoclasts, as gene expression data show enrichment for cytoskeleton organization in these cells.

In the CHO-SC co-culture experiments, phalloidin and FABP5 staining is actually strongest at the interface where mixing of the two cell populations occurs. Given that new data added to the revised manuscript attributes the FABP5 signal to SCs in these experiments (Supplementary Fig. 7c, d), strong phalloidin staining of the FABP5+ cells at the interface is consistent with our *in vivo* scRNA-seq data. Moreover, phalloidin staining in septoclasts *ex vivo* is strongly increased on calcium phosphate-treated culture surfaces (Supplementary Fig. 7e).

Taken together, the data indicates that the phenotype of SCs is strongly modulated by external signals such as the interaction with chondrocytes.

In addition, it is not clear what the localization is of the SCs after 5 days related to the chondrocytes and it seems that MMP9 is expressed by the chondrocytes and not by the SCs, although the authors state that MMP9 is expressed in SCs (line 178). Proper validation of the type of cell that produce MMP9, and other MMPs (MMP13, MMP14), is necessary, as chondrocytes can also increase MMP9 production in stressed conditions.

We have added new results attributing MMP expression to a specific cell type in the co-culture experiment.

Chondrocytes were isolated from *Acan-Cre-tdTomato* reporter mice. Following the co-culture period, chondrocytes (CHO) expressing tdTomato and tdTomato-negative SCs were separated by FACS. The isolated cell populations show the expected expression of markers – specifically, *Fabp5* in SCs and *Acan* in CHOs. SCs also show much higher expression of *Mmp9* and *Mmp14* transcripts, which is fully consistent with our findings *in vivo*.

The septoclasts seem to release some coated Ca-P, although this level seems very limited when you compare it to osteoclasts cultures in other studies. Further proof of the resorptive capacity of the SC should therefore be provided.

This experiment was solely conducted in response to questions raised during the review of the original submission and we would like to clarify once more that we do not claim that SCs play a substantial role in the resorption of calcified bone. In fact, SCs are concentrated near the growth plate and, unlike OCs, not on the surface of calcified bone. SCs also do not express vATPase, which is critical for bone resorption by osteoclasts. Instead, we argue that SCs are involved in cartilage resorption, which is supported by the data shown in our manuscript.

The following statement has been added to the Discussion: “*Unlike osteoclasts, SCs are not associated with calcified bone and show no expression of vATPase, which is critical for local pH changes allowing the dissolution of calcium phosphate. This argues that septoclasts are unlikely to mediate the resorption of calcified bone.*”

Reviewer #2 (Remarks to the Author):

The authors should be commended for having done a large amount of work to address the concerns of the reviewers. My own concerns are been addressed satisfactorily.

We thank the reviewer for this assessment and the useful suggestions during the first round of this review.

Reviewer #3 (Remarks to the Author):

General comments:

The authors have performed extensive additional analyses and included new images of improved quality to underline their statements. However, they replied to the comments and issues raised by the reviewers in an only partially satisfactory way. As a general comment, the reviewer would appreciate if the changes performed in the revised version could be highlighted for easier navigation through the manuscript and better appreciation of the performed additional analyses and their discussion.

Specific comments to authors responses to reviewer 3:

#1. As requested, additional staining for pericyte markers was performed and implemented as Supplemental Figure 2 d,e.

Thank you for this question.

#2. Although claimed to be “additional”, the revised Fig 2d seems identical to the previous Fig 2c. In addition, it remains unclear where the newly added single optical plane figure (from the rebuttal) can be found in the manuscript.

Thank you for pointing this out. The revised Fig. 1d is indeed identical with Fig. 1c in the original submission. The separated channels for this image were provided only in our response letter.

The question was whether some FABP5+ cells also express CD68/Vav. In our view, this is addressed by data shown in Fig. 2h, Fig. 3g, Suppl. Fig. 1e, Suppl. Fig. 5e, and Suppl. Fig. 6h. These images show that CD68+ or Vav-Cre-labelled cells are distinct from FABP5+ SCs. As we work with thick cryosections capturing tissue organization in 3D, pictures at lower resolution might give the impression that signals are occasionally co-localized even though they come from distinct cells located in different optical planes. However, images at higher magnification, such as the representative examples mentioned above, clearly support our findings and conclusions.

#3. As requested, the authors have performed additional ex vivo experiments with Dll4 now

clearly showing activation of notch signalling in SCs (Fig 4k). These results now support the authors' statement of the importance of notch signalling in septoclasts. Unlike claimed in the rebuttal, however, the content of sentence of the first submission (p11, 268: "Septoclasts are also controlled by Notch signalling through interactions with Dll4+ vessel buds at the chondro-osseous interface in the fracture callus") was not changed in the revised manuscript (p.12, 291 "...septoclasts are controlled by Notch signalling through interactions with Dll4+ vessel buds at the chondro-osseous interface").

Thank you for pointing this out. Contrary to what was stated in our response letter, this sentence was actually not changed. We now say "*Furthermore, we propose that septoclasts are controlled by Notch signalling through interactions with Dll4+ vessel buds at the chondro-osseous interface.*" This statement accurately reflects that the sentence it is a conclusion based on the data presented in the Results.

#4. The authors did not consider the comment of the reviewer that the aim of the study was rather generic: "...systematically investigate the processes during developmental and regenerative osteogenesis at cellular resolution" (p.3, line 61 in first submission). The stated aim is still identical in the revised version: "...systematically investigate the processes during developmental and regenerative osteogenesis at cellular resolution (p. 3, line 61). The reviewer asked to at least provide a rationale in the first sentence of the results, why instead of the proposed systematic investigation only a specific staining for FABP5 was performed. This was however not included. The authors only newly mention that they performed immunostaining "with a range of markers". However, the indicated figures (Fig 1a,b; Suppl Fig 1 a-c) still only show FABP5 staining.

We apologize that the reviewer is not fully satisfied by our statement in the Introduction. The sentence in question might appear somewhat generic but it is, in our view, nevertheless correct. It is in the nature of publications that much of the negative and therefore often less relevant data is not presented. At the same time, our study contains a substantial amount of scRNA-seq analysis, which is unbiased, systematic and allows, for example, the comparison of metalloproteinase expression in different cell types.

#5. The authors convincingly replied to the question on the timing of tamoxifen, however this should also be mentioned in the manuscript (as previously requested by the reviewer).

Thank you for pointing this out. We have now added the following sentence to the Discussion (bottom of page 12): "*This conclusion is also consistent with our previous characterization of Pdgfrb-CreERT2-expressing cells in developing bone, which showed that the transgene initially labels mesenchymal stromal cells in the metaphysis, which later give rise to other mesenchymal cell populations including reticular cells in marrow and adipocytes³⁵.*"

#6. Whether the authors have "...checked the wording of our text to ensure all our conclusions are covered by data" is difficult to judge, due to the lack of track changes/highlighted modification.

As part of this resubmission, we are providing a version of the manuscript in which all changes relative to the original submission are highlighted.

#7. The indicated grammatical mistake “from in” was not corrected, unlike claimed. (p. 4, line 68).

Thank you for alerting us to this oversight.

#9. “We have addressed this issue by providing more high magnification images and separated channels.” However, the original Fig. 2c seems identical to the newly claimed Fig. 2d (except the newly added square). See comment #2

The boxed area was added to indicate the region shown in the higher magnification panels on the right in Fig. 1d. As stated in our response to comment #2 (see above), we are providing several images, which convincingly show that FABP5+ cells are distinct from CD68+ or Vav-Cre-labelled cells.

#19. The newly included square in Fig. 2g should be mentioned in the legend. It is still rather misleading to see the overlay of the different stainings on the bottom left with the “heading”: GFP EMNC. This should be clarified (what is the bottom row?) in the figure and/or legend.

Agree. We have added the following sentence to the figure legend: “*Bottom panels show a single optical plane (left) and higher magnifications of the boxed area (centre and right), respectively (g).*”

#23. The newly added SafO definitely shows chondrocytes of a hypertrophic morphology. However, the COL X assessemtn that was asked for by the reviewer was not performed.

Agree. As already mentioned in our previous response letter, we were having some issues with the specificity of the Collagen type X immunostaining and are therefore not including any images in the revised manuscript. But even without these results it is clear that defective cartilage resorption is a major issue in EC-specific Dll4 mutants.

Additional comment(s):

Supplemental Figure: Please, indicate the mentioned area with “electron dense particles” with an arrow. What are the 3 different pictures? What is the difference between the left and the middle one? The right picture seems to be a higher magnification from the middle one. If so, please indicate the area (in the middle one). Also, the scale bar in the right figure seems to indicate a wrong scale, since it is identical to the one in the other figures, although showing higher magnification.

Thank you for pointing this out. We have added a red arrowhead to highlight the electron dense particle in the Supplemental Figure 2b and a red box indicating the area that is shown at higher magnification on the right. The labeling of the scale bars has been corrected and the figure legend has been extended: “*Electron micrographs of metaphysis region. SCs are located close to ECs and hypertrophic chondrocytes (HC) in the metaphysis (MP) (a). SCs physically interact with chondrocytes and contain electron dense particles (red arrowhead) (b). Area in the red box (centre) is shown at higher magnification on the right.*”

Supplemental Figure 2b

b

Reviewers' Comments:

Reviewer #1:

Remarks to the Author:

The authors have addressed the remaining concerns very adequately. I do not have further comments

Reviewer #3:

Remarks to the Author:

The authors have responded to all raised comments and questions and adapted the text accordingly where necessary. A couple of minor issues however remain, as detailed below:

ISSUE 1.

Reviewer: Although claimed to be "additional", the revised Fig 2d seems identical to the previous Fig 2c. In addition, it remains unclear where the newly added single optical plane figure (from the rebuttal) can be found in the manuscript.

Reply Authors: Thank you for pointing this out. The revised Fig. 1d is indeed identical with Fig. 1c in the original submission. The separated channels for this image were provided only in our response letter.

Reply Reviewer: Thanks for clarifying this. Separated channels, as shown for the reviewers, could still be added into the manuscript (e.g. in the supplemental), as these would also be of interest for the readers.

ISSUE 2.

Reviewer: The authors did not consider the comment of the reviewer that the aim of the study was rather generic: "...systematically investigate the processes during developmental and regenerative osteogenesis at cellular resolution" (p.3, line 61 in first submission). The stated aim is still identical in the revised version: "...systematically investigate the processes during developmental and regenerative osteogenesis at cellular resolution (p. 3, line 61). The reviewer asked to at least provide a rationale in the first sentence of the results, why instead of the proposed systematic investigation only a specific staining for FABP5 was performed. This was however not included. The authors only newly mention that they performed immunostaining "with a range of markers". However, the indicated figures (Fig 1a,b; Suppl Fig 1 a-c) still only show FABP5 staining.

Reply Authors: We apologize that the reviewer is not fully satisfied by our statement in the Introduction. The sentence in question might appear somewhat generic but it is, in our view, nevertheless correct. It is in the nature of publications that much of the negative and therefore often less relevant data is not presented. At the same time, our study contains a substantial amount of scRNA-seq analysis, which is unbiased, systematic and allows, for example, the comparison of metalloproteinase expression in different cell types.

Reply Reviewer: The authors are correct on the fact that, following an initial general approach, it is sound to focus on the investigation of a specific factor. However, it would be important to provide the rationale, WHY this specific factor, FABP5, was selected. The manuscript does not detail which was "the range of markers" that were assessed and why was FABP5 selected out of this (unknown) panel of markers.

REVIEWERS' COMMENTS

Reviewer #1 (Remarks to the Author):

The authors have addressed the remaining concerns very adequately. I do not have further comments

We appreciate your constructive suggestions, which have enabled us to improve the manuscript further.

Reviewer #3 (Remarks to the Author):

The authors have responded to all raised comments and questions and adapted the text accordingly where necessary. A couple of minor issues however remain, as detailed below:

ISSUE 1.

Reviewer: Although claimed to be “additional”, the revised Fig 2d seems identical to the previous Fig 2c. In addition, it remains unclear where the newly added single optical plane figure (from the rebuttal) can be found in the manuscript.

Reply Authors: Thank you for pointing this out. The revised Fig. 1d is indeed identical with Fig. 1c in the original submission. The separated channels for this image were provided only in our response letter.

Reply Reviewer: Thanks for clarifying this. Separated channels, as shown for the reviewers, could still be added into the manuscript (e.g. in the supplemental), as these would also be of interest for the readers.

Following the reviewer’s suggestion, we are now providing images showing the separated channels in Supplementary Figure 2i.

Supplementary Figure 2i

ISSUE 2.

Reviewer: The authors did not consider the comment of the reviewer that the aim of the study was rather generic: “...systematically investigate the processes during developmental and regenerative osteogenesis at cellular resolution” (p.3, line 61 in first submission). The stated

aim is still identical in the revised version: “..systematically investigate the processes during developmental and regenerative osteogenesis at cellular resolution (p. 3, line 61). The reviewer asked to at least provide a rationale in the first sentence of the results, why instead of the proposed systematic investigation only a specific staining for FABP5 was performed. This was however not included. The authors only newly mention that they performed immunostaining “with a range of markers”. However, the indicated figures (Fig 1a,b; Suppl Fig 1 a-c) still only show FABP5 staining.

Reply Authors: We apologize that the reviewer is not fully satisfied by our statement in the Introduction. The sentence in question might appear somewhat generic but it is, in our view, nevertheless correct. It is in the nature of publications that much of the negative and therefore often less relevant data is not presented. At the same time, our study contains a substantial amount of scRNA-seq analysis, which is unbiased, systematic and allows, for example, the comparison of metalloproteinase expression in different cell types.

Reply Reviewer: The authors are correct on the fact that, following an initial general approach, it is sound to focus on the investigation of a specific factor. However, it would be important to provide the rationale, WHY this specific factor, FABP5, was selected. The manuscript does not detail which was “the range of markers” that were assessed and why was FABP5 selected out of this (unknown) panel of markers.

The manuscript is actually showing many of these markers. We have looked at osteoclasts and endothelial cells, which have been previously linked to cartilage resorption. The Introduction also clearly states the rationale for looking at FABP5, which was identified as a septoclast marker by previous studies.